# Polyamide-based membranes with structural homogeneity for ultrafast molecular sieving

Liang Shen [1,2,8], Ruihuan Cheng [3,8], Ming Yi[1,2,8], Wei-Song Hung[4,5], Susilo Japip[6], Lian Tian[1,2], Xuan Zhang[1,2], Shudong Jiang[7], Song Li [3✉] & Yan Wang [1,2✉]

Thin-film composite membranes formed by conventional interfacial polymerization generally suffer from the depth heterogeneity of the polyamide layer, i.e., nonuniformly distributed free volume pores, leading to the inefficient permselectivity. Here, we demonstrate a facile and versatile approach to tune the nanoscale homogeneity of polyamide-based thin-film composite membranes via inorganic salt-mediated interfacial polymerization process. Molecular dynamics simulations and various characterization techniques elucidate in detail the underlying molecular mechanism by which the salt addition confines and regulates the diffusion of amine monomers to the water-oil interface and thus tunes the nanoscale homogeneity of the polyamide layer. The resulting thin-film composite membranes with thin, smooth, dense, and structurally homogeneous polyamide layers demonstrate a permeance increment of ~20–435% and/or solute rejection enhancement of ~10–170% as well as improved antifouling property for efficient reverse/forward osmosis and nanofiltration separations. This work sheds light on the tunability of the polyamide layer homogeneity via salt-regulated interfacial polymerization process.

[1] Key Laboratory of Material Chemistry for Energy Conversion and Storage (Huazhong University of Science and Technology), Ministry of Education, Wuhan 430074, China. [2] Hubei Key Laboratory of Material Chemistry and Service Failure, School of Chemistry and Chemical Engineering, Huazhong University of Science and Technology, Wuhan 430074, China. [3] School of Energy and Power Engineering, Huazhong University of Science and Technology, Wuhan 430074, China. [4] Graduate Institute of Applied Science and Technology, National Taiwan University of Science and Technology, Taipei 10607, Taiwan. [5] R&D Centre for Membrane Technology, Chung Yuan Christian University, Taoyuan 32023, Taiwan. [6] Department of Chemical & Biomolecular Engineering, National University of Singapore, 10 Kent Ridge Crescent, Singapore 119260, Singapore. [7] College of Chemistry and Chemical Engineering, Anhui University, 111 Jiulong Road, Hefei, Anhui 230601, China. [8] These authors contributed equally: Liang Shen, Ruihuan Cheng, Ming Yi. ✉email: songli@hust.edu.cn; wangyan@hust.edu.cn

The global population is estimated to increase by two billion by 2050[1]. The challenge of overcoming the water scarcity for anthropogenic activities with this burgeoning growth needs to be solved urgently[2]. Tremendous efforts have been devoted to exploiting water treatment technologies to produce clean water in recent decades[3,4]. For instance, membrane-based separation techniques exhibit transformative impacts on the implementation of desalination and water treatment at various levels[5–9], especially for polyamide (PA) based thin-film composite (TFC) membranes[10]. The fabrication by interfacial polymerization (IP) was first developed by Cadotte in the 1970s[11], and has been successfully commercialized in water purification via nanofiltration (NF), reverse osmosis (RO), and forward osmosis (FO) processes. The separation performance of a TFC membrane is primarily governed by the characteristics of the PA active layer, including its surface and bulk properties. The design and modification of the PA layer are thus of great importance to tune the separation performance. Extensive efforts have been devoted to this end, such as facilitating amine diffusion by adding surfactant or catalyst in monomer solutions[12–14], eliminating defective sites by a longer polymerization time or a higher reaction temperature[15–18], forming crumpled Turing structures[19,20], deployments of molecular layer-by-layer method[21] or electrospraying-assisted IP process[22], etc[23,24]. Despite tremendous advancements achieved via above strategies, researchers still cry out to the key target of bypassing the so-called permeability–selectivity trade-off to a greater extent.

To solve this nerve-wracking problem, deep insights should be applied to the complicated IP process to disclose the underlying mechanism. In principle, the PA selective layer of conventional TFC membranes is generally formed by polyamine monomers (typically m-phenylenediamine (MPD), piperazine (PIP), and polyethyleneimine (PEI)) in the aqueous phase and poly(acyl chloride) (typically trimesoyl chloride (TMC)) monomers in the organic phase at the water-oil interface via IP. PA layer growth is well accepted to be a diffusion-limited process[25], that is, PA layer formation is controlled by the diffusion of amine monomers to the reaction zone[26,27]. Due to the variation in diffusion resistance for amine monomers from the boundary interface to the reaction zone, the diffusivity and diffusion flux of amine monomer fluctuate, leading to spatially heterogeneous polymerization reaction[2]. As a result, the polymer density is highly nonuniformly distributed over the PA layer thickness[3], and the free volume pores in the PA layer are highly depth heterogeneous and widely distributed[28]. Previously, it was well-recognized that the nanoscale heterogeneity of the PA layer imposes positive impacts on the separation performance. However, 3D models of the nanoscale PA density maps of various reverse osmosis (RO) membranes constructed by Culp et al. revealed that the density fluctuations were detrimental to water transport[29,30]. They suggested that controlling over the internal nanoscale inhomogeneity could maximize the water permeance by minimizing mass fluctuations without sacrificing solute rejections. Therefore, improving the nanoscale homogeneity of the PA layer is presumably to be of great significance in surpassing the permeability-selectivity trade-off.

Several studies have been reported with respect to this topic. Liang et al. reported a surfactant assembly regulated IP to fabricate a PA layer with more uniform sub-nanometer pores, resulting from the faster and more homogeneous diffusion of amine monomers[28]. Similarly, precoating an intermediate layer of nanomaterials on the support layer before the IP process favored the controlled release of amine monomers to the water-oil interface, thus achieving an ultrathin and defect-free PA layer[20,31]. Additionally, a highly selective and permeable submicroporous TFC membrane with a large fraction of finely tuned structural submicroporosity was developed by using highly

contorted triptycene building blocks of bridged-bicyclic tetra-acyl chloride as monomers[32]. Recently, a combined IP and in situ sol-gel strategy was proposed to construct a PA layer with a narrow pore size distribution, where the space-limited hydrolysis and condensation of sol-gel effectively tuned the mean pore diameter and narrowed the pore size distribution[33]. However, a simple, low-cost, and effective, readily available method to tune the nanoscale homogeneity of TFC membranes for the co-improvement of permeability and selectivity is still preferred.

Herein, a facile and versatile approach is proposed to tune the nanoscale homogeneity of PA-based TFC membranes. In a conventional IP process, the obtained PA layer of TFC membranes intrinsically possesses multiscale heterogeneity[27], nonuniform free volume pores[28], high roughness, and thickness. To achieve smooth and thin PA layers with nanoscale homogeneity, the commonly used inorganic salts were added into the amine monomer solution during the IP process to confine and regulate the diffusion behavior of amine monomers toward the water-oil interface. Two types of inorganic salts were studied in this work, including neutral salt (taking NaCl as a typical example), and catalytic salt (taking $NaHCO_3$ as a typical example) that involves the removal of HCl byproducts generated by the IP reaction. Although previous studies have reported the use of salts for the fabrication of TFC membranes with more details summarized in Supplementary Table 1, they mainly focused on the effects of salt concentration on the chemical and morphological properties of the PA layer[34–39]. The nanoscale homogeneity influenced by the salt addition was still in the black box. Different from these studies, this work concentrates on elucidating the nanoscale homogeneity of the PA layer fabricated by the inorganic salt-regulated IP process, and extending such a strategy to various reaction systems, including MPD/TMC, PIP/TMC, PEI/TMC, and various salt types (i.e., LiCl, NaCl, KCl, $NH_4Cl$, $Na_2SO_4$, $NaHCO_3$, $Na_2CO_3$, $KHCO_3$, $NH_4HCO_3$, and $NaHSO_3$). The underlying mechanism for the tuned nanoscale homogeneity via the salt-regulated IP process was also disclosed by molecular dynamics (MD) simulations.

## Results and discussion

**Structural homogeneity of formed membranes.** The key to achieving nanoscale homogeneity of the PA layer is to tune the IP process, particularly for the diffusion behavior of amine monomers from the bulk aqueous phase to the reaction zone, which greatly impacts the formation of the dense PA film. In a conventional IP process as illustrated in Fig. 1a, the reaction between polyamine and poly(acyl chloride) occurs in the organic phase near the water-oil interface[40,41]. The formation of a dense PA layer consists of two stages: incipient film formation and film development in a diffusion-limited growth behavior[26]. According to a previous study[26], once the two kinds of reactive monomers mutually contact each other, a loose incipient film quickly forms. In this stage, the diffusivity of the amine monomer is governed by the resistance ($Z_0$) from the empty boundaries (without PA polymer) to the reaction zone. As the accumulation of PA polymer in the reaction zone filled in the loose incipient film, the diffusivity of amine monomer gradually decreases due to the steadily increased resistance of the incipient film ($Z_\delta$). When $Z_\delta$ approximate to $Z_0$, the IP process will slow down, resulting in the local rate of polymer accumulation being larger than the local rate of polymer formation. As a result, the middle part of the incipient film will be sealed by a thin and fairly uniform dense layer, which possesses the highest density constituting the selective barrier. Then, the IP process undergoes the well-known diffusion-limited growth regime ($Z_\delta > Z_0$). Over this regime, the middle layer in the incipient film gradually grows thicker and gives rise to a rougher

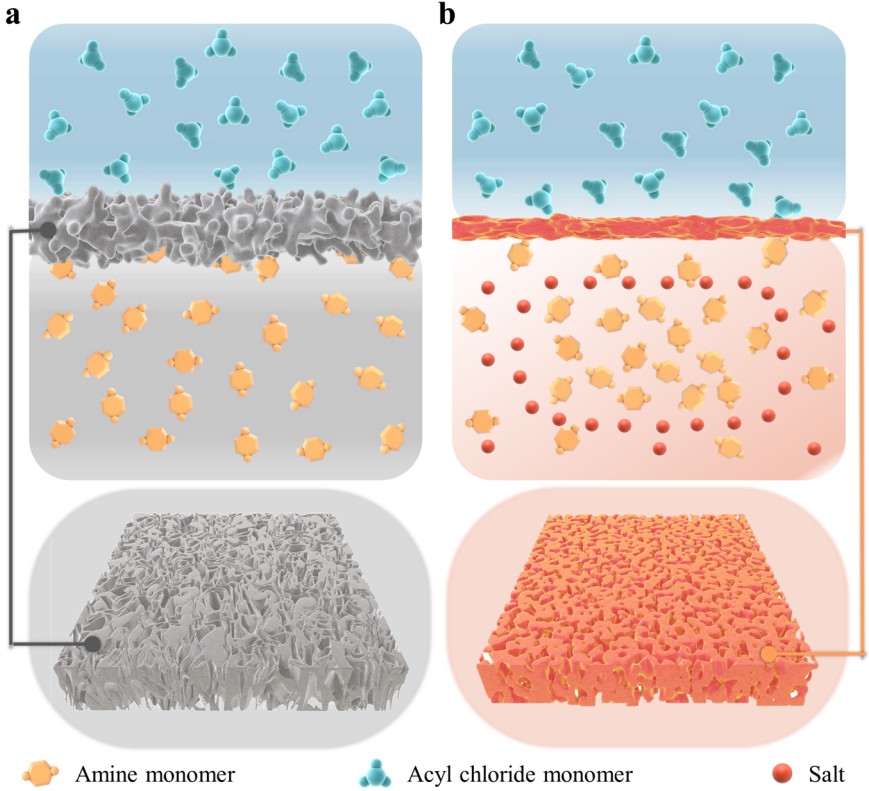

🟠 Amine monomer　　🔵 Acyl chloride monomer　　🔴 Salt

**Fig. 1 Schematical illustration of employed IP processes. (a)** conventional IP process, **(b)** salt-participating IP process.

morphology. Presumably, the final film is generally loose and heterogeneous in depth with a large void fraction[26,27].

Alternatively, the inorganic salt-tuned IP process occurs in a different manner, as depicted in Fig. 1b. On the one hand, the diffusivity of amine monomers toward the reaction zone is reduced in the presence of inorganic salts as later confirmed by MD simulations (see below). As a consequence, the local reaction rate accelerates and the reaction zone shrinks, causing a denser middle layer with thinner thickness buried inside the incipient loose film[26]. Unlike in the conventional IP process, the incipient film formed via the salt-participating IP process is less permeable for the diffusion of amine monomers, resulting in a faster slowdown. Subsequently, the growth of the incipient film in the final diffusion-limited growth regime is less significant, i.e., the less obvious morphological change and thickness increment.

On the other hand, salt ions can also be partially carried into the water-oil interface due to the interaction between cations and amine monomers[9,42]. Taking the amine monomer MPD and salt NaCl as examples, the interaction between MPD and $Na^+$ ions can be verified by the radial distribution function (see below). The profiles disclose that amine groups (N ($-NH_2$)–$Na^+$ pair) and benzene ring (benzene ring–$Na^+$ pair) of MPD tend to closely surround $Na^+$ ions compared to $H_2O$ (benzene ring–O ($H_2O$) and N ($-NH_2$)–O ($H_2O$) pairs) and $Cl^-$ ions (benzene ring--$Cl^-$ and N ($-NH_2$)–$Cl^-$ pairs), indicating the interactions between $Na^+$ ions and MPD molecules. Eventually, due to the Lewis acid-base complexation between cations (Lewis acid) and carbonyl groups (Lewis base) of the PA network, those salts trapped in the PA layer can be partially washed out (Supplementary Figs. 1, 2 and Supplementary Tables 2, 3). As a result, additional nanopores were generated in the modified PA layer as evidenced by the slightly larger free volume (Supplementary Fig. 3). Overall, the resultant PA layers fabricated by the salt-modified IP process are expected to be thin, dense, smooth, and structurally uniform with high porosity of free volume pores.

We first performed positron annihilation lifetime spectroscopy (PALS) characterization to examine the structural homogeneity of the resulting PA layers. Fig. 2a, b show that modified PA layers possess narrowly distributed free volume pore sizes, confirming our speculation of uniform dense layer. Instead, the pore size distribution of the pristine membrane formed via conventional IP is wider, suggesting the presence of heterogeneous free volume pores. This result is consistent with the rejection result of neutral organic solutes in Supplementary Fig. 4. Meanwhile, the in-depth O/N ratio profiles shown in Fig. 2c, Supplementary Fig. 5, and Supplementary Table 4 reveal that the O/N ratios of the modified membrane vary slightly against the detection depth in contrast to the fluctuating O/N ratio profiles of the pristine membrane, confirming the nanoscale homogeneity of salt-modified membranes again. Additionally, the O/N ratio profiles also substantiate our aforementioned speculation that the middle PA layer is relatively denser. Fig. 2a, b, and Supplementary Fig. 5 also show that the pore size of the modified PA layer is smaller than that of the pristine PA layer. Moreover, the data summarized in Supplementary Table 5 reveal that the modified PA layer is endowed with the higher free volume pore intensity ($I_3$) compared to that of the pristine membrane, agreeing well with our aforementioned hypothesis. The higher porosity is ascribed to the additional nanopores originating from the removal of trapped inorganic salts for the neutral salt-modified membranes, such as NaCl@MPD. Alternatively, the removal of trapped salts and nanobubbles of $CO_2$ generated by the reaction between catalytic salts and IP byproduct HCl[38] are responsible for the higher porosity of catalytic salt modified membranes (such as NaHCO$_3$@MPD), where the latter might contribute predominantly. It is noteworthy that despite the higher nanopore intensity in the modified PA layer, its fractional free volume ($FFV$) value (Supplementary Table 5) is still smaller than the $FFV$ of the pristine membrane, indicating the denser structure in agreement with our analysis above. This denser structure can also be verified

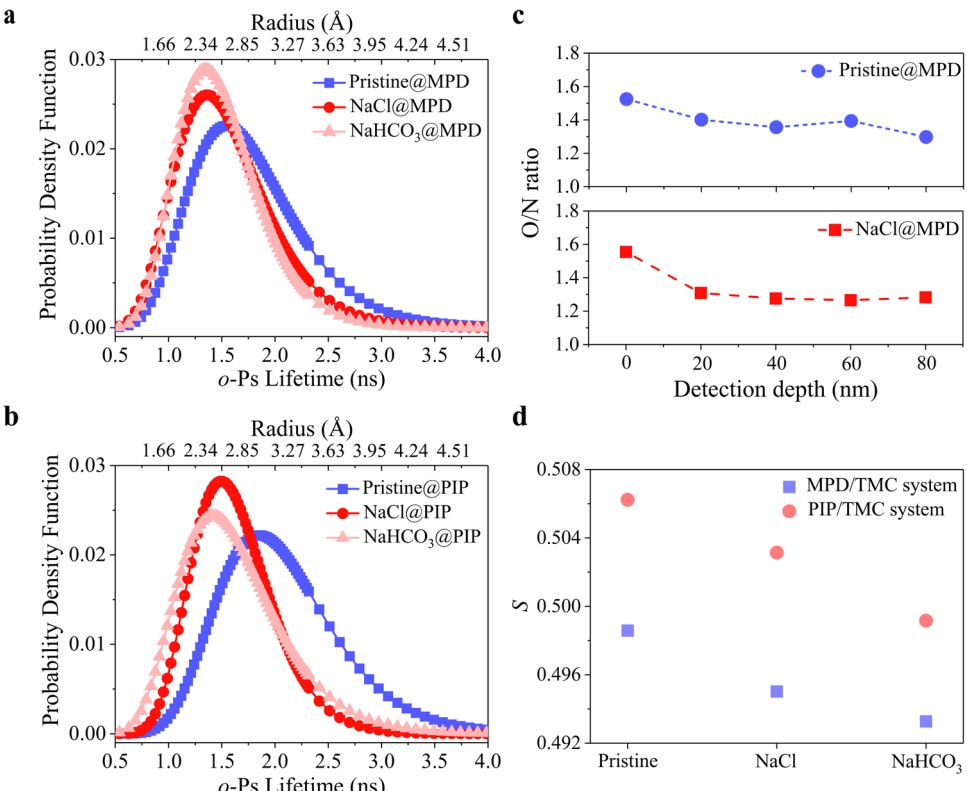

**Fig. 2 Microstructural properties of the pristine and modified membranes.** Free volume pore size distribution of the pristine and modified membranes formed by **(a)** MPD/TMC and **(b)** PIP/TMC. **(c)** In-depth O/N ratio profiles of membranes Pristine@MPD and NaCl@MPD. **(d)** $S$ values of the pristine and modified membranes for both MPD/TMC and PIP/TMC monomer systems.

by the smaller $S$ values, (Fig. 2d and Supplementary Fig. 6) smaller $d$-spacing values as determined by XRD results (Supplementary Fig. 7), and higher PA density (Supplementary Table 6).

**Molecular Dynamic simulation on both IP processes**. The generation of PA layer is a complex reaction-diffusion process. Given the consensus that amine monomers (taking MPD as an example) in the aqueous phase have to reach the water/hexane interface before reacting with TMC monomers in the organic phase, the inorganic salt in the aqueous phase may retard the diffusion of MPD monomers to the interface. To validate such an assumption, we performed MD simulations of the water/hexane interface in the presence of NaCl in the aqueous phase. The system without NaCl was also simulated as a control (Fig. 3a). It is found that MPD molecules in both simulation systems can diffuse from the bulk aqueous phase to the water/hexane interface. The $Na^+$ and $Cl^-$ ions tend to accumulate gradually and surround MPD molecules near the interface to retard their diffusion as shown in Fig. 3a and Supplementary Fig. 8a. In contrast, MPD molecules of the aqueous phase in the absence of NaCl are inclined to diffuse toward the interface.

To investigate the retarded diffusion manner of amine monomers, the mean square displacement (MSD) and self-diffusion coefficient ($D$) in the $z$-direction of MPD molecules that describes the motion of molecules were calculated. Fig. 3c shows that the MSD and self-diffusion coefficient of MPD in the system with NaCl ($D = 0.23 \times 10^{-9}\,m^2\,s^{-1}$) is nearly one-third of its counterpart in the system without NaCl ($D = 0.6 \times 10^{-9}\,m^2\,s^{-1}$), indicating the lower diffusivity of MPD molecules in the presence of NaCl. Such a difference in the self-diffusion coefficient results in

a smaller number of MPD molecules in the water/hexane interface in the NaCl-containing system, as confirmed by the number density ($\rho_N$) distribution of MPD (Fig. 3d and Supplementary Fig. 8b, c). According to the distribution of MPD in Fig. 3d, the distinct peak located at ~70 Å indicates the accumulation of amine monomers in the reaction zone. It can be found that the reaction zone width of the system with NaCl ($z = 63–78$ Å) is thinner than that of the system without NaCl ($z = 58–78$ Å). This shrunken reaction zone caused by the decreased diffusivity of MPD ultimately favors the thinner PA layer formed in the presence of NaCl. Additionally, the accumulation behavior of $Na^+$ and $Cl^-$ ions can be proven by the presence of two distinct peaks, leading to the aggregation of MPD monomers in the aqueous phase, as verified by the MPD peak located at 20-50 Å (Fig. 3d).

To examine the diffusive uniformity, we counted the number of MPD molecules diffused from the aqueous phase to the interface. Interestingly, the increased number of interfacial MPD molecules ($\Delta N$) was similar regardless of the presence of NaCl before ~2.5 ns (Fig. 3e), after which the $\Delta N$ of the system without NaCl was larger than the $\Delta N$ of the NaCl-containing system. Such a phenomenon demonstrates that a NaCl-enriched barrier was formed at ~2.5 ns in the presence of NaCl as shown in Supplementary Fig. 8a, retarding the diffusion of MPD molecules to the interface. Moreover, a more evident fluctuation in the $\Delta N$ of the system without NaCl was observed, indicating the nonuniform diffusive flux of MPD monomers. It results in spatially heterogeneous polymerization, thus leading to the formation of a widely distributed free volume pore size of the pristine PA layer[28]. In contrast, $\Delta N$ in the presence of NaCl steadily increases, indicating the enhanced uniformity of the MPD diffusive flux, which therefore contributes to the formation of structurally homogeneous PA layers[28] via the salt-participating

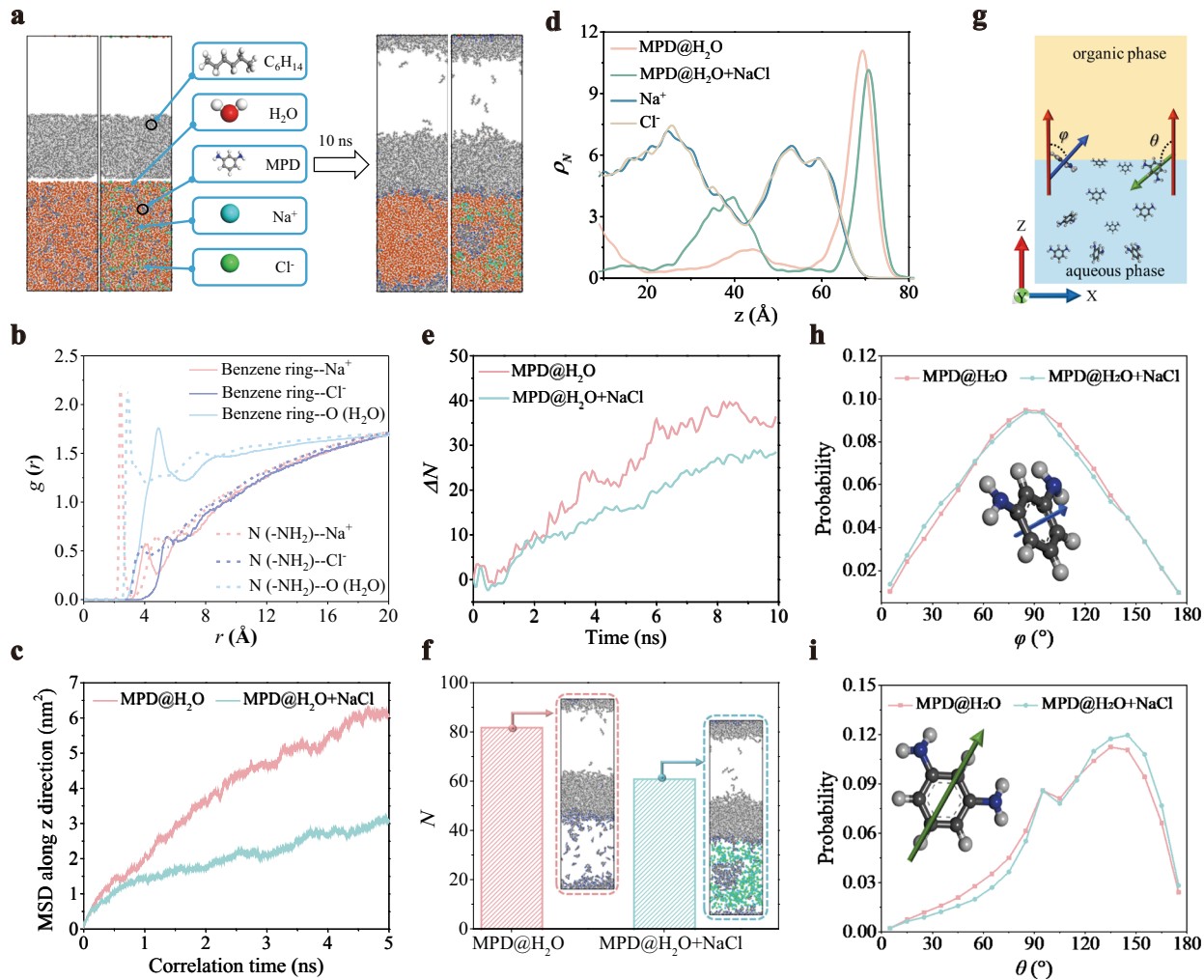

**Fig. 3 MD Simulation results. a** The initial (0 ns) and final (10 ns) snapshots of two systems with (right) and without (left) NaCl, respectively. **b** Radial distribution function of pairs MPD-$H_2O$ and MPD-NaCl: pairs of benzene ring–$Na^+$ (red solid line), benzene ring–$Cl^-$ (blue solid line), benzene ring–O (green solid line), N–$Na^+$ (red dot line), N–$Cl^-$ (blue dot line) and N–O (green dot line). **c** The MSD profiles of MPD molecules in two systems with and without NaCl, respectively. **d** The number density ($\rho_N$) distribution of MPD molecules (green and red lines), $Na^+$ (blue line), and $Cl^-$ (yellow line) ions in the two systems with and without NaCl, respectively. **e** The increased number variation of MPD molecules ($\Delta N = N - N_0$) compared with the initial value ($N_0$) and **f** the eventual number ($N$) of MPD at the interface of the two systems with and without NaCl, respectively. **g** Schematic diagram of the angles ($\varphi$ and $\theta$) formed between the positive direction of z-axis (red arrow, aqueous phase points to the organic phase) and the vectors perpendicular (blue) and parallel (green) to the benzene ring of MPD molecules. **(h)** The probability distribution of the angle ($\varphi$) formed between the positive direction of the z-axis and the vector perpendicular to the benzene ring of MPD at the interface of the systems with and without NaCl, respectively. **i** The probability distribution of the angle ($\theta$) formed between the positive direction of the z axis and the vector parallel to the benzene ring of MPD at the interface of the systems with and without NaCl, respectively.

IP process as testified by the PALS results (Fig. 2a, b). The eventual number of interfacial MPD molecules ($N$) in the presence of NaCl (~60) in the reaction zone is less than that in the system without NaCl (~80) (Fig. 3f), indicating a 25% reduction in the number of MPD molecules, which confirms the diffusion of MPD retarded by the presence of NaCl again.

To further explore the molecular behavior of the interfacial MPD molecules, the orientation of MPD molecules (as defined in Fig. 3g) at the interface with and without NaCl was analyzed (Fig. 3h, i). The benzene ring of MPD is prone to be parallel to the interface when $\varphi$ is closer to 0° or 180°, and the $\varphi$ distribution probabilities of MPD molecules in two systems are almost the same (Fig. 3h). The amine groups (i.e., −$NH_2$) of MPD are prone to orient toward the aqueous phase when $\theta > 90°$, and vice versa (Fig. 3i). However, there is a higher probability for MPD molecules to exhibit $\theta > 90°$ in the presence of NaCl, indicating

that the amine groups of MPD preferentially stay in the NaCl-containing aqueous phase, which is unfavorable for the reaction between MPD in the aqueous phase and TMC in the organic phase during IP process and thus benefits the formation of a thinner PA layer.

**Surface and morphology properties of formed membranes.** The restricted growth of the incipient PA film in the diffusion-limited growth regime and the decreased reaction zone thickness in the presence of NaCl favor the formation of thinner and smoother PA layers via the salt-participating IP process as confirmed by SEM, TEM, and AFM images (Fig. 4). Interestingly, the modified MPD-TMC PA layer exhibits regular honey-bomb-like Turing nanostructures rather than the coexistence of nodule- and leaf-like structures of the traditional PA layer (Fig. 4a). A closer observation of the surface morphology of modified membranes discloses that

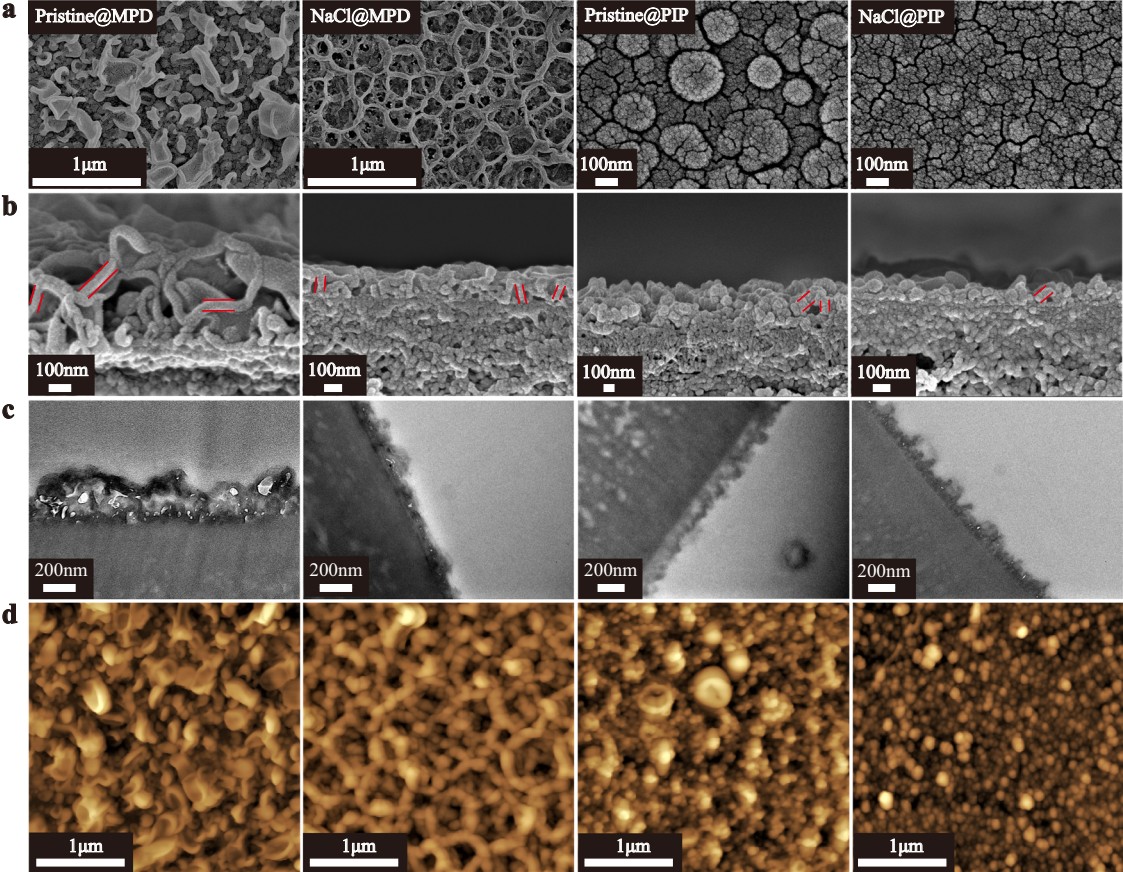

**Fig. 4 Morphology and topology of the pristine and modified membranes. a** Surface (scale bar: 1 μm and 100 nm) and **b** cross-sectional SEM images (scale bar: 100 nm), **c** cross-sectional TEM images (scale bar: 200 nm) and **d** topologic AFM images (scale bar: 1 μm) of the pristine and modified membranes.

the sunken thin PA films (crater) within the honey-bomb lattice (belt) cover the surface (Fig. 4a). This distinct belt-crater morphology might benefit water permeation as the belt creates additional filtration areas and the crater lowers the transport resistance. More specifically, Fig. 4b of cross-sectional images shows that both the pristine and modified membranes exhibit the copresence of small nodule-like and large leaf (pristine membrane)/belt (modified membrane)-like structures, where the large leaf/belt-like structure overlays the underneath nodular-like structure. Meanwhile, the cross-sectional TEM images in Fig. 4c show that the Pristine@MPD membrane exhibits both small discrete voids of nodule-like structures (lighter region) and large protuberances of leaf-like structures (darker region). Instead, belt-like structures cover the top surface of modified membranes (Fig. 4c). On the other side, the modified PIP-TMC PA layer displays smaller nodular structures compared to that of the pristine PA layer (Fig. 4a, b).

Inspection of cross-sectional SEM and TEM images reveals that the transparent thicknesses of the modified MPD-TMC and PIP-TMC PA layers show 62–67% and 44–59% thickness reduction (Fig. 4b, c, and Supplementary Figs. 9,10), respectively, as compared to their counterparts, correlating well with the MD simulation results. Moreover, the intrinsic thicknesses marked by red lines (PA wall) of modified membranes are even slightly thinner than that of the pristine membrane for both MPD/TMC and PIP/TMC monomer systems. The surface roughness of the PA layer has been reported to be correlated to its transparent thickness[26,27]. Accordingly, surface roughness decrements of 57–62% and 35–62% are achieved for modified MPD-TMC and PIP-TMC PA layers respectively compared to their counterparts (Fig. 4d and Supplementary Figs 9,10). Additionally, data

displayed in Supplementary Figs. 11, 12 reveal that modified PA layers are more hydrophilic and negatively charged than those of the pristine membrane, possibly due to the hydrolysis of acyl chloride groups caused by the complexion between cationic ions and carbonyl groups in TMC[34–36].

**Separation and antifouling properties of formed membranes.** These membranes fabricated by TMC and MPD were examined by FO and RO tests and the corresponding results are shown in Fig. 5a, b, and Supplementary Fig. 13. The water fluxes of the modified membranes in the FO process increase by ~56–86% (Fig. 5a) compared to that of the pristine membrane, resulting from the belt-crater morphology, thinner, and more hydrophilic PA layers with more water-permeable free volume pores. Except for the above merits, the improved nanoscale homogeneity could also be responsible for the improved water flux, which minimizes mass fluctuations to maximize the overall water permeability[29] as illustrated in Fig. 5c, d.

In addition, the reverse salt fluxes of modified membranes are lower than that of the pristine membrane, overcoming the permeability-selectivity tradeoff relationship. This behavior is due to the combined factors of increased negative charges and smaller free volume pore size. Additionally, the trapped salts in the PA layer repel solutes passing through the membrane, also favoring the improved solute rejection. It is noteworthy that NaHCO₃-modified membrane with a slightly narrower free volume pore size distribution shows a much lower reverse salt flux than that of NaCl-modified membrane, proving the significance of free volume pore uniformity on the separation performance. Consistent results were also obtained for RO tests. Fig. 5b shows that the water

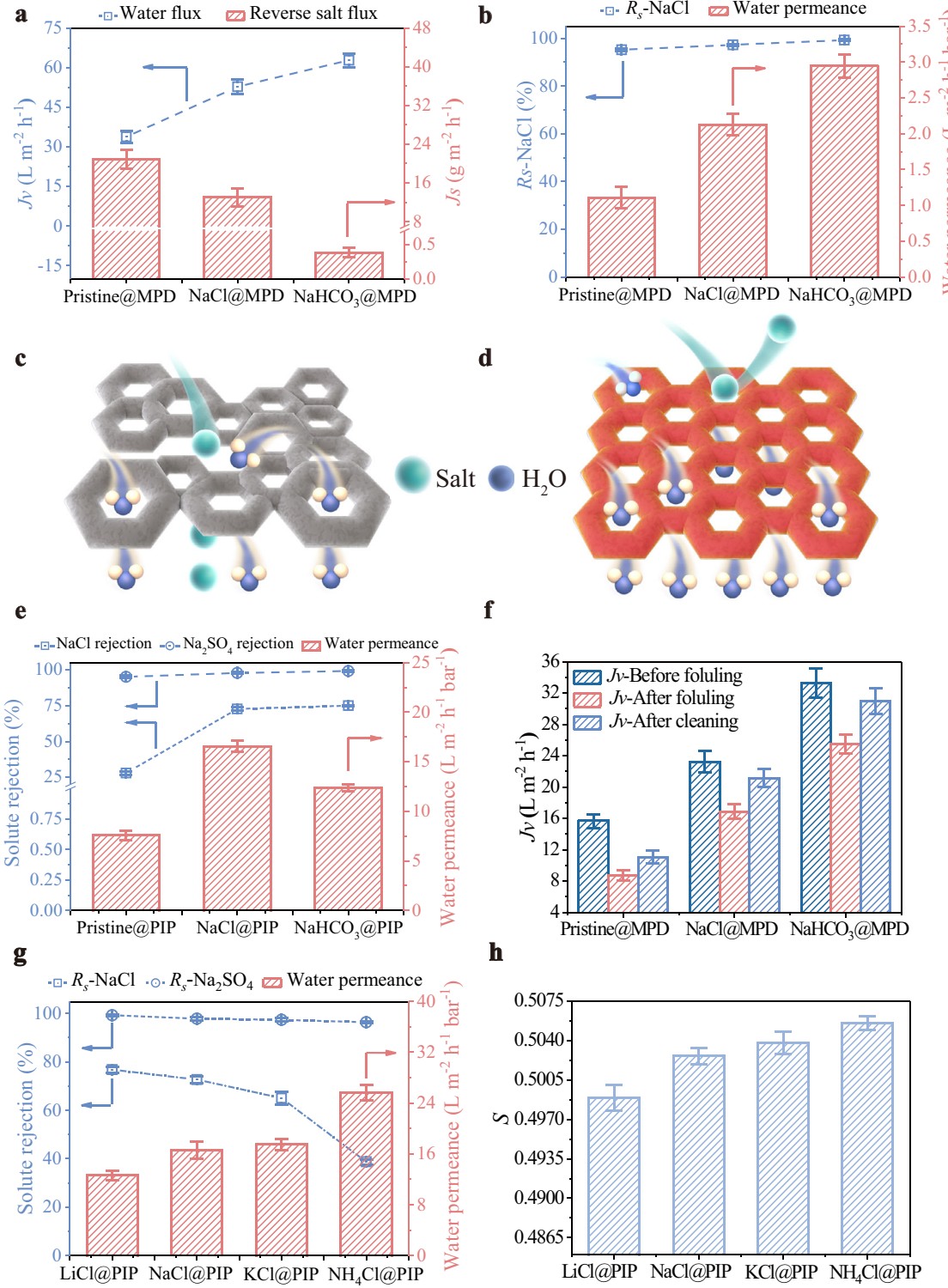

**Fig. 5 Separation and antifouling properties of the pristine and modified membranes. a** Water flux ($Jv$) and reverse salt flux ($Js$) of the pristine and modified membranes formed by MPD and TMC applied in FO process, under PRO mode using DI water and 2 M NaCl solution as the feed and draw solutions. **b** Water permeance ($A$) and NaCl rejections ($Rs$-NaCl) of the pristine and modified membranes formed by MPD and TMC using 1000 ppm NaCl solution as the feed applied in RO separation. Schematic illustrations about PA layers of **c** modified and **d** pristine membranes. **e** Water permeance ($A$) and solute rejections ($Rs$) of the pristine and modified membranes formed by PIP and TMC using 1000 ppm NaCl or Na$_2$SO$_4$ solution as the feed applied in NF process. **f** Dynamic fouling test results of the pristine and modified membranes formed by MPD and TMC in FO process. **g** Water permeance ($A$) and NaCl rejections ($Rs$-NaCl) of various salt-modified membranes formed by PIP and TMC using 1000 ppm NaCl or Na$_2$SO$_4$ solution as the feed applied in NF process. **h** $S$ values of various salt-modified membranes formed by PIP and TMC. The error bars represent the standard deviation and were calculated on the basis of at least three data points measured from different samples.

**Table 1 FO performance of TFC membranes formed by MPD/TMC modified with various salts and NF performance of TFC membranes formed by PIP/TMC modified with various salts (error bars represent standard deviation and were calculated on the basis of at least three data points measured from different samples).**

| Membrane code | $Jv^a$ (L m$^{-2}$ h$^{-1}$) (PRO/FO) | $Js^a$ (g m$^{-2}$ h$^{-1}$) (PRO/FO) | Membrane code | Water permeance$^b$ (L m$^{-2}$ h$^{-1}$ bar$^{-1}$) | Rejection (%)$^c$ NaCl/Na$_2$SO$_4$ |
|---|---|---|---|---|---|
| Pristine@MPD | 33.8 ± 2.2/15.7 ± 0.9 | 20.9 ± 1.9/11.4 ± 1.1 | Pristine@PIP | 7.6 ± 05 | 27.9 ± 1.0/95.1 ± 0.8 |
| LiCl@MPD | 42.9 ± 1.4/20.5 ± 0.8 | 11.2 ± 0.7/7.3 ± 0.6 | Na$_2$SO$_4$@PIP | 15.7 ± 0.7 | 60.3 ± 2.5/97.0 ± 0.3 |
| KCl@MPD | 53.8 ± 1.7/25.9 ± 0.1 | 13.1 ± 0.7/8.8 ± 0.2 | KHCO$_3$@PIP | 12.5 ± 0.1 | 66.6 ± 0.6/97.5 ± 0.4 |
| NH$_4$Cl@MPD | 65.2 ± 3.7/33.2 ± 2.4 | 30.1 ± 3.1/18.6 ± 2.3 | NH$_4$HCO$_3$@PIP | 13.5 ± 0.7 | 56.7 ± 1.3/96.8 ± 0.6 |
| KHCO$_3$@MPD | 69.5 ± 2.0/36.0 ± 1.1 | 23.0 ± 3.3/14.3 ± 2.5 | Na$_2$CO$_3$@PIP | 11.6 ± 0.2 | 62.8 ± 1.6/97.2 ± 0.6 |
| NH$_4$HCO$_3$@MPD | 66.2 ± 4.0/34.4 ± 3.2 | 12.9 ± 0.7/6.7 ± 0.7 | NaHSO$_3$@PIP | 38.9 ± 3.9 | 25.2 ± 2.4/94.9 ± 1.0 |

$^a$2 M NaCl solution and DI water were used as the draw and feed solutions respectively.
$^b$Ultrapure water was used as the feed, 5 bar of applied pressure.
$^c$1000 ppm NaCl or Na2SO4 solution was used as the feed, 5 bar of applied pressure.

permeances of modified membranes increase by ~92–165% while higher NaCl rejections are well maintained (~97–99% versus ~95%) compared to that of the pristine membrane.

On the other side, the separation performance of the TFC membrane fabricated by PIP and TMC was determined by NF tests. Fig. 5e shows that the water permeances of NaCl− and NaHCO$_3$-modified membranes increase by 119% and 63% respectively, compared to that of the pristine membrane. Meanwhile, solute (NaCl and Na$_2$SO$_4$) rejections of modified membranes increase compared to that of the pristine membrane. Especially for NaCl rejections of modified membranes, a significant improvement of ~160–169% is achieved compared to that of the pristine membrane. Based on the above results, we can deduce that the size-dependent exclusion behavior of PA-based membranes can be finely tuned by the uniformity of free volume pores.

For the practical application of membranes in water purification, the antifouling property is equally vital to the separation performance, which is closely related to the membrane span life. Numerous studies have revealed that membranes with hydrophilic and smooth surfaces favor a low fouling propensity[43,44]. The fouling resistance of the pristine and modified TFC membranes formed by TMC and MPD were also evaluated by dynamic fouling tests using synthetic wastewater containing sodium alginate (SA) and Ca$^{2+}$ ions as the foulant. The results shown in Fig. 5f and Supplementary Fig. 14 reveal that the water flux of the pristine membrane drops 44% and only recovers 14% (recovery ratio ~70%) after physical cleaning. Instead, the flux drops of modified membranes are only 23–27%, while the recovery ratio can reach ~91–93%, which is ascribed to the above two factors.

Apart from NaCl and NaHCO$_3$, other inorganic salts were also employed to examine their effects on the membrane properties and performance, including LiCl, KCl, NH$_4$Cl, Na$_2$SO$_4$, Na$_2$CO$_3$, KHCO$_3$, NH$_4$HCO$_3$, and NaHSO$_3$. Similar to NaCl and NaHCO$_3$, these salt-modified membranes also show smoother and thinner PA layers (Supplementary Fig. 15) with higher hydrophilicity (Supplementary Fig. 16).

Supplementary Fig. 17 and Table 1 show that the separation performance of modified membranes is affected by the type of inorganic salt. The salt with the larger ionic radius (Supplementary Table 7) is added, and the modified membrane shows the higher water permeance (Fig. 5g), possibly due to the removal of the larger salt that leaves the larger free volume pores in the PA layer, as confirmed by the larger $S$ value (Fig. 5h and Supplementary Fig. 18). However, the MD simulation results in Supplementary Fig. 19 suggest that the reduction in amine monomer diffusivity by adding inorganic salts with large sizes (such as NH$_4$Cl) is not as significant as that by adding NaCl

(Supplementary Fig. 19a) due to the weak salt effect of accumulation behavior (Supplementary Fig. 19b). As a result, NH$_4$Cl@PIP membrane shows a relatively less homogenous PA layer, as confirmed by the relatively wide pore size distribution (Supplementary Fig. 19c, d) compared to that of NaCl@PIP membrane. Therefore, we can deduce that the free volume and nanoscale homogeneity of the PA layer can be finely tuned by the selection of the inorganic salt, thus influencing the separation performance. It also can be seen that the water permeances of NH$_4$Cl− and NaHSO$_3$− modified NF membranes (PIP/TMC system) increase by 253% and 435% (Fig. 5g and Table 1) with the higher or comparable NaCl rejections maintained compared to that of the pristine membrane.

A comparison of our membranes with state-of-the-art membranes, including the laboratory-made and commercial membranes, is provided in Fig. 6a, b, and Supplementary Tables 8–11. Our membranes formed either by MPD/TMC or PIP/TMC present the higher water flux/permeance and higher/competitive salt rejections (or lower reverse salt flux) than previously reported laboratory-made and commercial FO/NF membranes. These reported strategies in previous studies mainly aimed at reducing the PA layer thickness, tailoring the intrinsic property or increasing the effective permeable area of the PA layer. Among them, the third strategy that achieves a rough crumple structure is proven to be of great effectiveness to improve the water permeability without sacrificing solute rejection. For a comparison, TFC membranes developed in this work not only show competitive separation performances, but also possess improved fouling resistance rather than the compromised antifouling capacity of rough structures. Therefore, the salt-tuned IP process is believed to be a promising platform for the fabrication of next-generation PA-based TFC membranes.

Alternatively, the addition of salt is effective not only in MPD/TMC and PIP/TMC systems, but also in PEI/TMC system. Supplementary Fig. 20 shows that in comparison with the pristine membrane, salt-modified membranes show higher water permeances (36–282% increment) and higher (7–36% improvement) or comparable NaCl rejections. Therefore, this facile and versatile modification approach for the IP process is proven to be effective in fabricating PA-based TFC membranes with high separation performances that surpass the permeability-selectivity trade-off.

Overall, we demonstrated a facile and versatile approach to tune the nanoscale homogeneity of PA-based TFC membranes via a salt-regulated IP process for various monomer systems and salt types to improve separation performance. The behavior of inorganic salts accumulated near the water-oil interface confines and regulates the diffusion of amine monomers into the reaction zone, resulting in the uniform diffusive flux of amine monomers, thus the spatially homogeneous polymerization and the

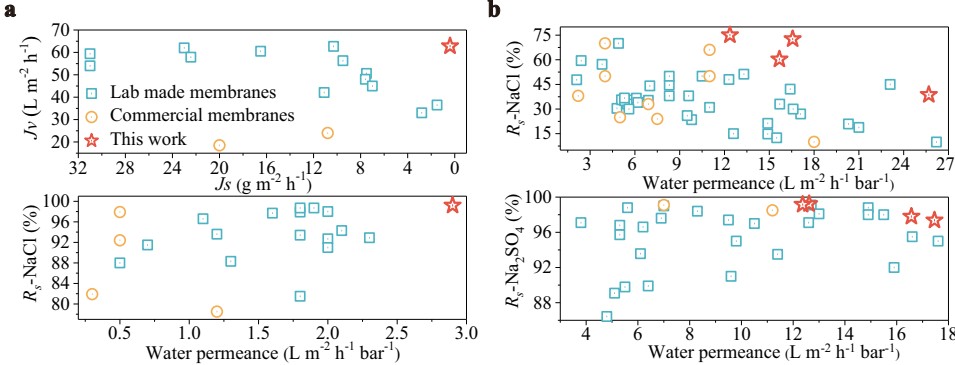

**Fig. 6 Separation performance benchmarking. a** Performance comparison with lab-made FO membranes (formed by MPD and TMC) and commercial FO membranes (upper) water flux versus reverse salt flux using 2 M NaCl solution and DI water as draw and feed solutions under PRO mode, and (bottom) intrinsic separation properties of water permeance and NaCl rejection. **b** Performance comparison with lab-made NF membranes (formed by PIP and TMC) and commercial FO membranes (upper) water permeance versus NaCl rejection, and (bottom) water permeance versus Na$_2$SO$_4$ rejection.

formation of smooth and thin PA layers with structural homogeneity. Additionally, trapped inorganic salts after removal and/or nanobubbles generated by the reaction between catalytic salts and byproduct HCl leave additional nanopores in the PA layer. As a result, the PA layer shows a higher porosity, and smaller and more uniformly distributed pore size. The resulting membranes, therefore, show greatly improved water permeance and/or solute rejection for all three monomer systems in FO/RO/NF separations. Additionally, the modified TFC membranes with smoother and more hydrophilic surfaces also show a lower fouling propensity. Moreover, the water permeance and solute rejection of the resulting membranes can be reasonably tuned by selecting different inorganic salts. Therefore, this work sheds insights into the fabrication of TFC membranes with structurally homogeneous PA selective layers for molecular sieving.

## Methods

**Preparation of TFC membranes.** The TFC membrane comprising the porous PSF substrate membrane and the dense PA layer was fabricated via nonsolvent induced phase inversion and IP methods respectively. The detailed fabrication parameters are listed in Supplementary Table 12. In brief, the degassed dope solution was poured on a clean glass plate and cast by a casting knife. Immediately, the membrane was transferred into a water bath to complete the phase separation. Next, the substrate membrane was stored in water to remove residual solvent. Then, the fresh substrate membrane was rinsed with ultrapure water and immersed in the amine solution for a predetermined time, and the excess amine solution on the substrate top surface was then removed by a rubber roller. Thereafter, the organic solution was brought in to contact the amine-saturated substrate surface, and then the organic solution was drained off to obtain the TFC membrane. Finally, these membranes were post-treated in an 80 °C water bath for 15 min to eliminate the salt crystallization and then stored in DI water before use. The salt concentration added to the aqueous phase was optimized based on the separation performance, as shown in Supplementary Fig. 21. The resulting membranes with and without modification for MPD-TMC, PIP-TMC, and PEI-TMC systems were denoted as Pristine@MPD/PIP/PEI such as (Pristine@MPD) and Salt@MPD/PIP/PEI (such as NaCl@MPD or NaCl@PIP), respectively.

**Characterizations.** The chemical compositions of the PA layer were examined by X-ray Photoelectron Spectroscopy (XPS, AXIS UltraDLD, Kratos, England) with a monochromatic Al Kα X-ray source. The C$_{24}$H$_{12}^+$ ion source was used for depth profiling operated at 12 keV energy with an etching rate of 15 nm/min calibrated by Ta$_2$O$_5$. The surface hydrophilicity and surface streaming potentials of the PA layers were measured by contact angle goniometer (DSA 25, KRÜSS, Germany) and zeta potential analyzer (SurPASS™ 3, Anton Paar, Austria). The morphology and topology of the PA layers were observed by Field Emission Scanning Electron Microscopy (FESEM, SU8010, Hitachi, Japan) and Atomic Force Microscope (AFM, SPM9700, Shimadzu, Japan). The cross-sectional structure was observed by Transmission Electron Microscopy (TEM, JEM-F200, JEOL, Japan) at an accelerating voltage of 100 kV. The fresh membrane sample was immersed in SPI resin and followed by sectioning using a Leica EM UC7 ultramicrotome. The microstructures of the PA layers were examined by X-ray Diffraction (XRD, Smart-Lab, Rigku, Japan) and Positron Annihilation Lifetime Spectra (PALS, Chung Yuan University). The elemental distribution in the PA layer was examined by Energy

Disperse Spectroscopy (EDS, SU8010, Hitachi, Japan). The trapped Na content in the TFC membrane and free-standing PA film were detected by Atomic Absorption Spectrum (AAS, iCE 3000, Thermo Scientific, USA). The skeletal density of the free-standing PA films was measured by a pycnometer (AccuPyc 1330, Micromeritics, USA) utilizing Helium as the probing gas.

**Determination of pore size distribution.** The pore size distribution of TFC membranes formed by PIP and TMC was determined by the rejection test using 200 ppm feed solution containing neutral organic solutes (i.e., glycerol (92 Da), glucose (180 Da), sucrose (342 Da), and PEG600 (600 Da)) under 5 bar. The detailed experimental procedures are described below[28,33]. Briefly, the permeate sample of a fresh membrane sample was collected after stabilization at 5 bar for 1 h. Next, the solute concentrations in both the feed and the permeate were measured by a Total Organic Carbon (TOC) analyzer (vario TOC, Elementar, German).

**Separation performance evaluation.** The separation performance of fabricated membranes was evaluated by osmotic-driven and pressure-driven tests at room temperature. Briefly for osmotic-driven tests, DI water and 2 M NaCl applied as the feed and draw solutions respectively were recycled to the FO system by two peristaltic pumps at a flow rate of 0.3 L/min. Each fresh membrane sample was subjected to two operation modes: active layering facing draw solution (FO mode) and active layer facing feed solution (pressure retarded osmosis, PRO mode). All data were collected after stabilization (30 min) for each fresh membrane sample and at least three parallel tests were repeated. The pressure-driven tests were conducted using a crossflow apparatus (Suzhou Faith Hope Membrane Technology) for RO and NF separations. In brief, the fresh membrane samples were stabilized at 6 bar for 1 h before data collection. The water permeance (A) was evaluated using pure water as the feed, and the solute rejection ($R_s$) was measured with 1000 ppm NaCl or Na$_2$SO$_4$ solution as the feed, under an applied pressure of 5 bar.

The osmotic-driven separation performance was evaluated by the water flux ($J_v$) and reverse salt flux ($J_s$) determined by Eqs. (1) and (2),

$$J_v = \frac{\Delta V_o}{A_{m,o}\Delta t} \tag{1}$$

$$J_s = \frac{\Delta (C_t V_t)}{A_{m,o}\Delta t} \tag{2}$$

where $\Delta V_o$ is the volume change of the draw solution over the testing time $\Delta t$ monitored by a digital balance (FX3000-GD, AND, Japan), $A_{m,o}$ is the membrane area (3.9 cm$^2$), $C_t$ and $V_t$ are the salt concentration and volume of the feed solution detected by a conductivity meter (FE30, Mettler Toledo, Switzerland), respectively.

The pressure-driven separation performance was evaluated by water permeance (A) and solute rejection ($R_s$) based on Eqs. (3) and (4),

$$A = \frac{\Delta V_p}{A_{m,p} \times \Delta t \times \Delta P} \tag{3}$$

$$R_S = \left(1 - \frac{C_p}{C_f}\right) \times 100\% \tag{4}$$

where $\Delta V_p$ is the permeate volume change over the testing time, $A_{m,p}$ is the effective membrane area (7.1 cm$^2$), $\Delta P$ is the transmembrane pressure, $C_p$ and $C_f$ are the solute concentrations of the permeate and the feed detected by a conductivity meter (FE30, Mettler Toledo, Switzerland).

**Dynamic fouling tests**. The dynamic fouling tests were conducted by a FO setup under FO mode at room temperature. In brief, a fresh membrane sample was preconditioned by DI water (as both the feed and draw solutions) for 1 h. Then, a synthetic wastewater solution without SA foulants (0.45 mM KH$_2$PO$_4$, 9.20 mM NaCl, 0.61 mM MgSO$_4$, 0.5 mM NaHCO$_3$, 0.5 mM CaCl$_2$, and 0.93 mM NH$_4$Cl)[43] and a 2 M NaCl solution replaced the DI water as the feed and draw solutions respectively to obtain the initial water flux for 1 h. Thereafter, the synthetic wastewater containing SA was used as the feed solution to perform the 18-h fouling test. Then, the fouled membrane was physically cleaned for 30 min by increasing the flow rate from 0.3 to 0.6 L min$^{-1}$ using DI water as both the feed and draw solutions. Ultimately, the recovered water flux of the cleaned membrane sample was measured again using DI water and 2 M NaCl solution as the feed and draw solutions respectively.

**Molecular Dynamics simulation**. As shown in Fig. 2a, two simulation boxes $(50 \times 50 \times 180 \text{ Å}^3)$ were constructed to simulate a water-hexane interface with the same amount of water and hexane molecules, in which one box is filled with both MPD and NaCl dissolved in water and the other is only filled with MPD and water molecules. Both systems were comprised of the same number of hexane (500), water (4800), and MPD (200) molecules. In the system with NaCl dissolved in water, there were 320 Na$^+$ and Cl$^-$ ions randomly dissolved in water. The force field (FF) parameters for organic molecules (i.e., hexane and MPD) were OPLS-AA[45]. Water and NaCl were modeled using the SPC potential[46] and SD potential[47], respectively. The initial configurations of these two systems were optimized by PACKMOL[48]. Then, we adopted the large-scale atomic/molecular massively parallel simulator (LAMMPS)[49] package to simulate these two systems for 10.0 ns in NVT ensemble at 300 K. The time step of the simulations was 1.0 fs. The last 1.0 ns was recorded for data analysis. The MSD in the *z-direction* of MPD molecules that describes the motion of molecules was calculated according to $MSD = \langle |r(t) - r(0)|^2 \rangle$, in which $r(t)$ and $r(0)$ represent the position of a particle at time $t$ and the initial time, respectively, and $\langle \rangle$ represents the ensemble average. Accordingly, the self-diffusion coefficient ($D$) of MPD can be derived from MSD based on $D = \lim_{t \to \infty} \frac{1}{2t} \langle |r(t) - r(0)|^2 \rangle$.

## Data availability

Source data are provided with this paper.

## Code availability

The codes for simulations performed in this study (MD) are provided with this paper.

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

## Acknowledgements
This work was supported by the National Key Research and Development Program of China (no. 2020YFB1709301) and Natural Science Foundation of Hubei Scientific Committee (no. 2016CFA001). We also appreciate Prof. Tai-Shung Chung at the National University of Singapore for PALS characterization.

## Author contributions
L.S. and Y.W. designed the experiments. L.S., M.Y., X.Z., W-S.H., S.J. and L.T. performed the experiments and characterizations. R-H.C. and S.L. performed molecular dynamics (MD) simulation. All co-authors discussed the results. L.S., M.Y., R-H.C., S-D.J., Y.W. and S.L. wrote and revised the manuscript.

## Competing interests
The authors declare no competing interests.
