## [Peer Review File · Nature Communications]

REVIEWER COMMENTS

Reviewer #1 (Remarks to the Author):

317322: Polyamide-based membranes with structural homogeneity for
2 ultrafast molecular sieving

This paper reported the addition of various salts as aqueous amine solution additives in the interfacial polymerization, and investigated the salt effect in the fabrication of thin-film composite membrane. The work is very comprehensive, particularly in MD part. Authors have great amount of theoretical work added to explain the salt effect during interfacial polymerization. However, I am afraid there is not enough novelty for me to recommend to be published in this journal, as explained in the following bullet points 1&2. Other special membrane field journals, such as Journal of Membrane Science or Desalination may be more suitable.

1. The concept of adding salt in aqueous amine solution has long history, almost from beginning of making commercial TFC membrane. In several earlier published US patents, people used amine salts as pore formed agents or moiety keeper to avoid pore collapse during the fabrication of TFC membrane, they observed major flux increase over pristine TFC membrane.

2. As recently and cited in this paper (Ma, X.-H. et al. Nanofoaming of polyamide desalination membranes to tune permeability and selectivity. Environmental Science & Technology Letters 5,123-130 (2018). CY Tang group used NaHCO₃ during interfacial polymerization to form the nanofoam by byproduct CO₂, and found greatly flux increase.

3. Other question in this paper's content:

152: On the other hand, salt ions can also be partially carried into the water-oil interface due to the coordination interaction between the cationic ions and the amine groups of the amine monomer. Eventually, almost trapped salts in the formed PA layer can be washed out, as confirmed by the reduced salt concentration in the washing water (Figure S1) and the decreased atom percentage examined by EDX.

Question & Comments: How these coordination interaction happened? Author should explain, particularly for Na ion with amine groups, and also, if salt is trapped in nanopore, it may difficult to remove.

158 : Overall, the resultant PA layers fabricated by the salt-modified IP process are expected to be thin, dense, smooth and structurally uniform with high porosity of free volume pore.

Question & Comments: Based on the Zeta data, this membrane are very negative charged in nature, is there enough long chain polyamide nanofilm formed?

171: The higher porosity is ascribed to the additional nanopores originated from the removal of trapped inorganic salts and/or nanobubbles of CO₂ generated by reaction between NaHCO₃ and IP byproduct HCl

Question & Comments: These two factors contribute the higher porosity, author may need to separate them and find out which one is more dominated

187 : It is found that both MPD and TMC molecules can diffuse from the bulk aqueous phase to the water/hexane interface

Question & Comments: How TMC diffuse from the bulk aqueous phase to the water/hexane interface?

273: It can be found in Figure 4b that the water permeances of modified membranes increase by ~92-165% while higher NaCl rejections well maintained (~97-99% versus ~95%) as compared to that of the pristine membrane.

Question & Comments: This rejection increase could just be the results of the increase of surface negative charges, judged by Zeta results.

276: On the other side, the separation performance of TFC membrane fabricated by PIP and TMC was determined by NF tests. Figure 4c shows that water permeances of NaCl- and NaHCO₃-modified membranes increase by 119% and 63% respectively compared to that of the pristine membrane. Meanwhile, solute (NaCl and Na₂SO₄) rejections of modified membranes increase compared to that of the pristine membrane. Especially for NaCl rejections of modified membranes, the significant improvement of ~160-169% is achieved compared to that of the pristine membrane.

Question & Comments: The rejection of salt for PIP based membrane is normally below 40%, like pristine@PIP, but NaCl@PIP and NaHCO₃@PIP is unusually high; in fact, these membranes become loose RO membrane made by MPD, which is not good, judged by the normal function of NF membrane: separation of NaCl from Na₂SO₄. Therefore, I guess because there is a lot of temperately trapped NaCl, it repels the NaCl and other salts pass through membrane, which slows down the dissolve/diffuse process during normal separation

Reviewer #2 (Remarks to the Author):

This study used a facile and versatile modification on IP process by adding different inorganic salts into the aqueous monomer solution to fabricate TFC membranes with depth-homogeneous PA layer. Molecular dynamics simulation and various characterization techniques fully demonstrated that the salt addition confines and regulates the diffusion of amine monomers to the water-oil interface and thus tunes the nanoscale-homogeneity of the PA layer. The properties of the prepared TFC membrane were improved, such as permeation flux, solute rejection and anti-fouling performance. This report is interesting and worthy to publish after a major revising. However, for the benefit of the readers and publishers, some issue should be explained and modified before being accepted. They are as followed:

1. In Lines of 145-147, "As a result,, causing a denser middle layer with thinner thickness" . In Lines of 255-258, "XPS results summarized in Table S3 and Figure 1g, indicating the lower crosslinking degree of the modified PA layers". How does the author explain the relationship between a denser middle layer with thinner thickness and the reduced crosslinking degree of the modified PA layer?

2. There are some mistakes in the marking of Figure 4(b), please correct them.

3. The contrast of (top) SEM images of modified membranes before washing in Figure S2 (a) are too small to be clear. Please place the clearer images.

4. Based on the results of Figure S1, Figure S2 and Table S1, how can the author determine that the Na atom percentage of the modified membrane after washing is all the salts trapped in the formed PA layer rather than deposited on the PSF substrate or between the substrate and PA layer?

5. In the Section of Method, "these membranes annealed at 80 for 15 min" . As known, soluble inorganic salts can be thermally induced crystallization. Did the author consider the effect of the generation of inorganic salts crystal on PA layer in the heat treatment process and hence affects the morphology of PA layer, or even the formation of large crystals at high inorganic salt concentration, resulting in defects of PA layer.

6. There are some tiny grammar mistakes in this manuscript draft. Please re-read it carefully and

correct them.

Reviewer #3 (Remarks to the Author):

The manuscript presents an interesting and simple way of modifying the interfacial polymerization process by adding inorganic salts into the aqueous phase. The authors report improvements in performance of the formed membranes. I think the manuscript would be acceptable for publication following a major revision as noted below:

1) The English language needs serious improvements – the manuscript is, at times, difficult to read. For example: “this result is consistent to the rejection”, “IP process undergoes a different manner”, “As the result, the locate rate of reaction accelerates”, “As the result, the resulting PA layer” and many, many other language problems which make the reception of the manuscript very difficult.

2) In the introduction, while discussing Figure 1a, the Authors seem to suggest that generally the IP layers of TFC membranes have low salt rejections. However, commercial RO membranes have salt rejections upwards of 99% even for monovalent salts. Therefore such statements are misleading to the reader in my opinion. It is the low water permeance that is the key problem that has to be addressed or an improvement of the balance between rejection and water permeance.

3) The authors say that: “scarce studies have been reported ... to finely tuning the pore size of free volume pores” in the literature. However, they fail to cite the recent work which was dedicated precisely to addressing these issues:

a. <https://doi.org/10.1002/adma.202001132>

b. <https://doi.org/10.1016/j.memsci.2020.118572>

c. <https://patents.google.com/patent/US20190299168A1/en>

d. <https://doi.org/10.1016/j.memsci.2019.02.032>

e. <https://doi.org/10.1039/C7TA07819F>

4) I am also missing the acknowledgment that it is currently well-established in the field that heterogeneity of the IP layer structure in the TFC membranes has a significant positive impact on its properties. It is not true that flat (low roughness) and homogeneous layers are hard to achieve (see PIP-based commercial membranes) or that they have better separation properties. Very rough structures of tight RO membranes also perform excellent in real applications. The Authors make it sound as if achieving a homogenous membrane is the Holy Grail. This is not true. In fact, achievement of a relatively thick dense layer (50 – 100 nm) is often less beneficial than highly rough crumpled structure with effectively very thin skins of the “bubbles” (~10-20 nm).

5) Figure 4b has a swapped legend between the permeance and rejection.

6) I am missing information on how experimentally was the PALS done – as far as I am aware, PALS need a penetration depth on the scale of millimeters. Was the TFC IP layer isolated and stacked? Or was it done on powder in a “one-pot” reaction – that is not on an actual selective layer?

7) How significant really is the MPD distribution difference with and without NaCl derived from MD? It seems to me that a difference between 63-78 A and 58-78 A zones would not make a huge difference. Also, what does “partial MPD monomers in aqueous phase” mean?

8) There are very little details given in the SI regarding the performance tests.

9) Looking at tables S6, S7, S8 it seems to me that the separation performance in the RO and NF of the presented membranes is not drastically different to the multitude of membranes prepared in the literature, with maybe the exception of NF with NaCl.

10) How was the thickness of the IP layers determined? I cannot find any information on that in the main manuscript or in the SI. SEM –based determination is usually very difficult in TFC composites. Did the Authors attempt to isolate the layer and measure the thickness after deposition on a silicon wafer with AFM? Or with other methods?

Point-by-point responses to reviewers

on

Polyamide-based membranes with structural homogeneity for ultrafast molecular sieving

(Manuscript ID: NCOMMS-21-23772)

We thank the editor and reviewers for their careful review and constructive comments and suggestions on the manuscript. We have performed additional experiments and revised the manuscript accordingly, which, we believe has strengthened the manuscript. We hope that the revised manuscript meets the standard for publication. A point-to-point response to reviewer comments is provided below.

Reviewer #1 (Remarks to the Author):

317322: Polyamide-based membranes with structural homogeneity for ultrafast molecular sieving

This paper reported the addition of various salts as aqueous amine solution additives in the interfacial polymerization, and investigated the salt effect in the fabrication of thin-film composite membrane. The work is very comprehensive, particularly in MD part. Authors have great amount of theoretical work added to explain the salt effect during interfacial polymerization. However, I am afraid there is not enough novelty for me to recommend to be published in this journal, as explained in the following bullet points 1&2. Other special membrane field journals, such as Journal of Membrane Science or Desalination may be more suitable.

Reply:

Thank you for your positive comments and helpful suggestions on this work. **For the novelty, we would like to emphasize that our work elucidated the nanoscale homogeneity of polyamide-based TFC membranes fabricated via inorganic salt-tuned interfacial polymerization and disclosed the underlying molecular mechanism for the first time.** We believe this topic has not been covered in previous studies using salts for the fabrication of TFC membranes, and we have provided the point-by-point responses and the relevant revisions in the manuscript below.

Moreover, to strength our work, we supplemented some more experiments and characterizations as shown in **Figures R1-R4**. The in-depth O/N ratio profiles shown in **Figure R1-b** indicate that the PA network of NaCl@MPD membrane is relatively more uniform than that of Pristine@MPD membrane, further confirming the nanoscale homogeneity of salt-modified membranes. Additionally, TEM images of both the pristine and modified membranes were taken to exhibit the cross-sectional morphology and get a more accurate thickness of the PA layer as shown in **Figure R2**. Cross-sectional SEM images with higher resolution of as-fabricated TFC membranes were also retaken as shown in **Figure R3**. It shows that both the transparent thickness and intrinsic thickness (PA wall marked by red lines) of modified membrane are thinner than that of the pristine membrane. Moreover, results in **Figure R4** reveal that the deployment of salt with large radius size (such as NH₄Cl) causes less reduction in amine monomer diffusivity (**Figure R4-a**) due to the weak salt effect of accumulation behavior (**Figure R4-b**). As a result, NH₄Cl@PIP membrane shows the relatively less homogenous PA layer as confirmed by the relatively wider pore size distribution (**Figure R4-d**) compared to that of NaCl@PIP membrane.

Details are given in the point-by-point reply as follows.

Figure R1 XPS depth profiles of membranes Pristine@MPD and NaCl@MPD: **(a)** wide-scan spectra, and **(b)** O/N ratios.

Figure R2 TEM images of the pristine and modified membranes

Figure R3 SEM images of the pristine and modified membranes

Figure R4 (a) MSD profiles of MPD molecules in systems of pure MPD, MPD+NaCl, and MPD+NH₄Cl. **(b)** The number density (ρ_N) distribution of MPD molecules in the two systems of pure MPD, MPD+NaCl, and MPD+NH₄Cl. **(c)** Solute rejection as a function of solute molecular weight and **(d)** pore size distribution of membranes Pristine@PIP, NaCl@PIP, NH₄Cl@PIP.

1. The concept of adding salt in aqueous amine solution has long history, almost from beginning of making commercial TFC membrane. In several earlier published US patents, people used amine salts as pore formed agents or moiety keeper to avoid pore collapse during the fabrication of TFC membrane, they observed major flux increase over pristine TFC membrane.

Reply:

Thank you for your valuable comment.

We agree with you that the concept of adding salts in aqueous amine solution has long history according to published patents and papers. Nevertheless, this work is another story. As mentioned above, **we reported an effective approach and the underlying molecular mechanism to tune the nanoscale homogeneity of polyamide-based TFC membranes via inorganic salt-regulated interfacial polymerization for improved separation performance in this work.** In order to compare with other studies, we have summarized related patents and literature works in **Table R1**.

Table R1 Summary of reported salt additives added in aqueous amine solution for fabricating TFC membranes

Additive category	Additives	Aqueous monomer	Key findings about salt effects	Refs.
Organic salts	TEBAB/TMBAB/ TEBAC	MPD/ PIP	1) Phase transfer catalysts improved the polymerization efficiency in interfacial polymerization by helping the monomer in the water phase move into the organic layer.	[1]
	CSA-TEA	MPD	1) TEA acted as a catalyst to accelerate the MPD–TMC reaction by neutralizing HCl produced during amide formation. 2) Addition of CSA protected the microporous skin layer of the support membrane from annealing during curing.	[2]
	TEA+SDS+BMIC/ OMIC	PIP	1) OMIC had a large polar head and a long hydrophobic tail in the structure, playing the role of a surfactant. 2) BMIC as a phase transfer catalyst, not only improved the transfer mechanism across the interface, but also increased the diffusion rate.	[3]
	SDS+CTAB	PIP	1) The porosity of thin layers decreased in the presence of CTAB and SDS. 2) Surfactant-modified membranes showed a dense and compressed thin layer.	[4]
	TEAC/TBAB/ CSA-TEA/BMMIC	PIP	1) Amine salt containing larger steric configuration cationic amine group resulted in a TFC membrane with better performance. 2) The extent of cross-linking was increased with increasing molecular weight of	[5]

			cationic amine groups.	
	TEAC/TBAB/ CSA-TEA/BTMAC/ BTEAC/DTMAC	PIP	 1) A phase transfer additive should had suitable organic structures and be loose enough to ensure high capacity to absorb anionic or complex agent and acted as a phase transfer catalyst. 2) The ammonium salt with more and longer lipophilic alkyl groups showed the higher catalytic efficiency. [6] 3) The larger ammonium salt complex would take over larger space in polymer, and result in larger free volume. 	
	Surfactant + amine salt	Amine	The salt effect was not stated.	[7-9]
	Amine salt	Amine	The salt effect was not stated.	[10]
	Surfactant + amine salt	Amine	The salt effect was not stated.	[11]
	Tertiary amine salt	Amine	1) Tertiary amine act as both pore forming agent and catalyst (absorbing the IP reaction by-products).	[12-14]
Inorganic salts	LiBr + SDS (Alcohol amine)	TEOA/ MDEOA	 1) The surface roughness and coverage varied with the variation of LiBr concentration. 2) Li⁺ ions induce an increased interaction coefficient between TMC and TEOA, forming a dense skin layer. [15] 3) Complexation between Li⁺ ion and the carbonyl in TMC causes the hydrolysis 	

of acid chloride groups of TMC, inducing a hydrophilic and loose surface layer.

CaCl ₂	TEPA	 1) The addition of CaCl₂ enhanced the interfacial tension and reduced the mass transfer of TEPA to organic phase. 2) Higher CaCl₂ content induced an unevenly distributed interfacial polymerization reaction, resulting in the increased surface roughness. 	[16]
TEA+NaCl (Nanofiber support)	PIP	 1) The addition of salt to aqueous phase increased the interfacial tension, depressing the mass transfer of PIP and constructing a loose PA layer. 2) Further increase NaCl content induced the uneven distribution in nanoscale of the IP reaction, leading to the morphological structure transition from the crisscrossed ridge networks to crowded nodular arrays. 	[17]
NaHCO ₃	MPD	 1) Under different pH conditions, different amount of CO₂ nanobubbles generated resulting different morphological features and separation properties 2) Under different concentration conditions, the resulting membranes showed different morphological features, thus tuning the roughness and separation properties 	[18, 19]
NaCl (Heat treatment)	PIP	 1) The growth of inorganic salt crystals stretched the nascent and flexible PA layer, and the inorganic salt crystals can sacrifice themselves in water, contributing to the formation of a thin and rough PA layer with crumpled nanostructure. 	[20]

2) The density and size of NaCl crystals increase with the increasing NaCl concentration, resulting in unidirectional scattering distribution to interconnected dendritic distribution;

It can be seen that both organic and inorganic salts were added in the amine solution in previous studies, but organic salts are much more commonly used. Generally, these organic salts act as (a) phase transfer catalyst or surfactant, which aims to facilitate the diffusion of amine monomer to the reaction zone; (b) catalyst that neutralizes HCl byproduct to complete IP reaction; (c) pore-forming agent (e.g. amine salts), IP reaction; (c) pore-forming agent (e.g. amine salts), which increases the free volume of the PA layer after its removal; (d) moiety keeper to avoid pore collapse during the fabrication of TFC membrane. **However, the functions of organic salts on the IP behavior and the formation of PA layer are quite different to that of the inorganic salt, i.e., the addition of organic salts facilitates the diffusion of amine monomers toward the organic phase.** Additionally, membranes developed in this work show superior separation performance to those membranes making with organic salts.

For these studies introducing inorganic salts (such as LiCl, CaCl₂, NaCl, NaHCO₃) into the aqueous amine solution to fabricate TFC membranes, **they mainly aimed at the effects of salt concentration on the chemical and morphological properties of membranes.** For example, the higher CaCl₂ or NaCl content induced an unevenly distributed interfacial polymerization reaction, resulting in the morphological transition, or the higher NaHCO₃ content resulted in the morphology translated from leave-nodule morphology to belt-crater morphology then ruptured roughness structure. **However, these works failed to investigate the underlying influences and mechanism of the salt addition on the nanoscale homogeneity of the formed PA layer via tuning the IP behavior.**

Different from above works, we focused on exploring the nanoscale homogeneity of the PA layer fabricated via inorganic salt-tuned IP process by MD simulations

and various characterizations for improved separation performance. Based on simulation and experimental characterizations, the new findings were summarized in this work as follows:

- 1) The presence of inorganic salts benefited the **uniform diffusion flux of amine monomers (Figure 5R-a)**, contributing to the spatially homogeneous polymerization and **the formation of structurally homogeneous PA layers (Figure R5-b and c)**.
- 2) The simulation results revealed that the addition of inorganic salt in the aqueous caused a decreased diffusivity of amine monomers to the reaction zone (**Figure R5-d**) **by an accumulation behavior of salts surrounding amine monomers (Figure R5-e)**, resulting in a decreased reaction zone thickness (**Figure R5-f**) and thus the formation of thinner, smoother and denser PA layers (**Figure R5-g and h**).

As a consequence, the modified membranes showed an improved permselectivity. Specifically, merits of higher porosity, thinner PA layer, better nanoscale homogeneity and hydrophilicity were responsible for the higher water permeance. Besides, the smaller pore size, more negative charges and repulsion from trapped salts in the PA layer contributed to higher solute rejections.

Additionally, **the universal application of salt-modified IP process can be extended to various reaction systems** (MPD/TMC, PIP/TMC, PEI/TMC) **and salt types** (LiCl, NaCl, KCl, NH₄Cl, Na₂SO₄, NaHCO₃, Na₂CO₃, KHCO₃, NH₄HCO₃, NaHSO₃). **The free volume property also can be intentionally tuned by the selection of inorganic salt**, thus optimizing the separation properties of resulting membranes.

With the discussion given above, this work is of great significance to provide an

experimental and theoretical guidance on how to achieve the nanoscale homogeneity *via* the salt-tuned IP process for improved separation performance. This work also provides insightful guidance for the selection of inorganic salts to tune the free volume properties of the PA layer and corresponding separation properties.

Corresponding revision has been in the revised manuscript as follows:

“Extensive efforts have been devoted to this end, such as facilitating amine diffusion by adding surfactant or catalyst in monomer solutions¹²⁻¹⁴, eliminating defective sites by a longer polymerization time or a higher reaction temperature¹⁵⁻¹⁸, forming crumpled Turing structures^{19,20}, deployments of molecular layer-by-layer method²¹ or electro spraying-assisted IP process²², etc^{23,24}.” (Page 3)

“Although previous studies have reported the use of salts for the fabrication of TFC membranes with more details summarized in **Table S1**, they mainly focused on the effects of salt concentration on the chemical and morphological properties of the PA layer³⁴⁻³⁹. The nanoscale homogeneity influenced by the salt addition was still in the black box. Different from these studies, this work concentrates on elucidating the nanoscale homogeneity of the PA layer fabricated by the inorganic salt-regulated IP process, and extending such a strategy to various reaction systems, including m-phenyldiamine (MPD)/TMC, piperazine (PIP)/TMC, polyethyleneimine (PEI)/TMC, and various salt types (i.e., LiCl, NaCl, KCl, NH₄Cl, Na₂SO₄, NaHCO₃, Na₂CO₃, KHCO₃, NH₄HCO₃, NaHSO₃). The underlying mechanism for the tuned nanoscale homogeneity *via* the salt-regulated IP process was also disclosed by molecular dynamics simulations.” (Pages 5-6)

Figure R5 New findings discovered in this work. **(a)** The increased number variation of MPD molecules ($\Delta N = N - N_0$) compared with the initial value (N_0). **(b)** Free volume pore size distribution of the pristine and modified membranes formed by PIP and TMC. **(c)** In-depth profiles of O/N ratios of membranes Pristine@MPD and NaCl@MPD by XPS analysis. **(d)** The mean squared displacement (MSD) profiles of MPD molecules in two systems with and without NaCl, respectively. **(e)** Simulation box showing the accumulation behavior of NaCl surrounded MPD monomers in the aqueous phase. **(f)** The number density (ρ_N) distribution of MPD molecules (green and yellow lines), Na^+ (blue line) and Cl^- (red line) ions in the two systems with and without NaCl, respectively. **(g)** SEM images of membranes Pristine@MPD and NaCl@MPD. **(h)** S values of the pristine and modified membranes for both MPD/TMC and PIP/TMC monomer systems.

2. As recently and cited in this paper (Ma, X.-H. et al. Nanofoaming of polyamide desalination membranes to tune permeability and selectivity. Environmental Science & Technology Letters 5,123-130 (2018). CY Tang group used NaHCO₃ during interfacial polymerization to form the nanofoam by byproduct CO₂, and found greatly flux increase.

Reply:

Thank you for your comment.

In the cited work, Prof. Tang added NaHCO₃ into the amine solution to fabricate TFC membranes to **investigate the effect of produced CO₂ nanobubbles on the formed PA layer.** By adjusting the pH value of amine solution to vary the carbonate chemistry, the production of CO₂ can be controlled, thus tuning the morphology, structure and separation properties of resultant TFC membranes. However, **the influences of NaHCO₃ in the aqueous solution on the diffusion behavior of amine monomers during IP process and the homogeneity of formed PA layer were not taken into consideration in their work.**

Instead, as stated above, our present work **comprehensively investigates the diffusion behavior of amine monomers in the presence of inorganic salt that able to tune the nanoscale homogeneity of the formed PA layers.** Additionally, extended studies on various monomer systems and salt types were also conducted. This could be meaningful guidance for the selection of inorganic salt for the fabrication of TFC membranes.

In a word, **the research content and target** of our present work are therefore quite different from those of Prof. Tang's work.

3. Other question in this paper's content:

152: On the other hand, salt ions can also be partially carried into the water-oil interface due to the coordination interaction between the cationic ions and the amine groups of the amine monomer. Eventually, almost trapped salts in the formed PA layer can be washed out, as confirmed by the reduced salt concentration in the washing water (Figure S1) and the decreased atom percentage examined by EDX.

Question & Comments: How these coordination interactions happened? Author should explain, particularly for Na ion with amine groups, and also, if salt is trapped in nanopore, it may difficult to remove.

Reply:

Thank you for your valuable questions.

Actually, there are two kinds of interaction between MPD and Na⁺ ions. The first one is the cation- π interactions between Na⁺ ions and benzene ring groups in MPD molecules due to the presence of empty orbitals of the cation and the delocalized π states of the benzene ring structure[21, 22]. Second, the amine groups of MPD molecules also interact with Na⁺ ions due to the lone pair of electrons of the nitrogen atoms in the amine group[22]. The above speculation can be verified by the radial distribution function profiles in **Figure R6**, which is a statistical result based on aforementioned MD simulation. It can be found that Na⁺ ions tend to closely surround the amine groups and benzene ring in MPD compared to H₂O and Cl⁻ ions, indicating the interaction between Na⁺ ions and MPD.

Figure R6 Radial distribution function of pairs benzene ring--Na⁺ (red solid line), benzene ring--Cl⁻ (blue solid line), benzene ring—O (H₂O) (green solid line), N (-NH₂)--Na⁺ (red dot line), N (-NH₂)--Cl⁻ (blue dot line) and N (-NH₂)--O (H₂O) (green dot line).

Besides, we agree with you that it is difficult to remove them from the membrane completely, especially those trapped in the PA layer, as identified by EDX results as shown in **Figure R7**. This is due to the Lewis acid-base complexation between the cation ions (Lewis acid) and carbonyl groups (Lewis base) in the PA layer[15-17, 23, 24]. Additionally, atomic absorption spectroscopy (AAS, iCE 3000, Thermo Scientific, USA) was performed for both the salt-modified TFC membranes (10*6 cm²) and free-standing PA layers (70 mg), which were washed by DI water for 1 day and 6 days. It can be found that the salt content trapped in the PA layer after 1-day and 6-day washing decreases slightly from 0.0902/0.0888 to 0.0871/0.0856 μg/cm² of membranes NaCl@MPD/NaHCO₃@MPD, respectively. On the other side, the content of trapped salt in the free-standing PA layer (NaCl@MPD) reduces slightly from

161.455 mg/kg to 157.280 mg/kg for 1-day and 6-day washing. The above results are consistent with the result displayed in **Figure R8**. It can be deduced that the large amount of salt removed in the first day is generally the salt trapped in the substrate membrane as well as between the substrate membrane and the PA layer. However, after the first-day washing, the salt trapped in the PA layer can be partially washed out with longer washing duration (6 days) due to the complexation between the salt ions and the carbonyl groups of the PA network^{15-17,23,24}. The ultimate salt content trapped accounts for 0.016% of the weight of the PA layer.

Corresponding revision has been made in the revised manuscript as follows:

“On the other hand, salt ions can also be partially carried into the water-oil interface due to the interaction between cations and amine monomers^{9,42}. Taking the amine monomer MPD and salt NaCl as examples, the interaction between MPD and Na⁺ ions can be verified by the radial distribution function in **Figure 2b**. The profiles disclose that amine groups (N (-NH₂)-Na⁺ pair) and benzene ring (benzene ring--Na⁺ pair) of MPD tend to closely surround Na⁺ ions compared to H₂O (benzene ring--O (H₂O) and N (-NH₂)-O (H₂O) pairs) and Cl⁻ ions (benzene ring--Cl⁻ and N (-NH₂)-Cl⁻ pairs), indicating the interactions between Na⁺ ions and MPD molecules. Eventually, due to the Lewis acid-base complexation between cations (Lewis acid) and carbonyl groups (Lewis base) of the PA network, those salts trapped in the PA layer can be partially washed out (**Figures S1-S2** and **Tables S2-S3**). As a result, additional nanopores were generated in the modified PA layer as evidenced by the slightly larger free volume (**Figure S3**).” (**Pages 7-8**)

Figure R7 Atom percent of Na in salt modified membranes before and after washing.

Figure R8 NaCl concentration in the washing water of the MPD@NaCl membrane

158: Overall, the resultant PA layers fabricated by the salt-modified IP process are expected to be thin, dense, smooth and structurally uniform with high porosity of free volume pore.

Question & Comments: Based on the Zeta data, this membrane are very negative charged in nature, is there enough long chain polyamide nanofilm formed?

Reply:

Thank you for your valuable comment.

The increased surface negative charge of modified membranes is due to the increased hydrolysis propensity of acyl chloride groups in TMC caused by the complexation between metal ions (Na^+) and carbonyl groups in TMC^{15-17,23,24}. Compared to TFC membranes reported in literatures[18, 25], O/N ratios of our modified membranes are rational, suggesting the successful formation of PA nanofilms with efficient crosslinking and therefore long polyamide chains formed. Additionally, the in-depth profiles in **Figure R1** reveal that the O/N ratios of the modified membrane in deeper positions are even slightly lower than that of the pristine membrane, suggesting the formation of a relatively denser middle layer with a higher crosslinking degree.

Besides, **Figure R9** shows that both free-standing PA films can be obtained in the reaction system with and without NaCl, demonstrating the successful formation of PA layer with the presence of NaCl in the amine solution. This also can be evidenced by the characteristic peaks at 1660 (stretch vibration of carbonyl group) and 1545 cm^{-1} (coupling of the in-plane N–H bending and C–N stretching vibrations) in FTIR spectra as shown in **Figure R10**.

On the other side, TGA (STA449F3, NETZSCH, German) and DSC (DSC2500, TA, USA) analysis were performed for free standing PA films (Pristine@MPD and NaCl@MPD) formed with and without NaCl after washing 6 days. The DSC and TGA results displayed in **Figure R11** reveal that both T_g and decomposition temperature (T_d) of NaCl@MPD are approximate to those of Pristine@MPD. Moreover, we also have measured the density of the free-standing PA films by a pycnometer (AccuPyc 1330, Micromeritics, USA) utilizing Helium as the probing gas. The density of NaCl@MPD ($1.238 \pm 0.023 \text{ g cm}^{-3}$) is slightly higher than that of Pristine@MPD ($1.183 \pm 0.017 \text{ g cm}^{-3}$). Presumably, the molecular weight of the PA nanofilm formed with salt addition should be approximate to the that of the pristine

PA nanofilm.

Therefore, we can deduce from above results that there is enough long chain polyamide nanofilm formed with the presence of salt during the IP process.

Figure R1 XPS depth profiles of membranes Pristine@MPD and NaCl@MPD: (a) wide-scan spectra, and (b) O/N ratios.

Figure R9 Photo of self-standing PA film formed by TMC (0.15 w/v in hexane) and MPD 3.4 wt% in H₂O). (Left: without NaCl, right: with 3 wt% NaCl)

Figure R10 ATR-FTIR spectra of the pristine and modified membrane formed by MPD and TMC

Figure R11 (a) DSC and **(b)** TGA results of free-standing PA films of Pristine@MPD and NaCl@MPD.

171: The higher porosity is ascribed to the additional nanopores originated from the removal of trapped inorganic salts and/or nanobubbles of CO₂ generated by reaction between NaHCO₃ and IP byproduct HCl.

Question & Comments: These two factors contribute the higher porosity, author may

need to separate them and find out which one is more dominated.

Reply:

Thank you for your valuable suggestion.

The higher porosity of the neutral salt (NaCl) modified membrane is solely resulted from the removal of trapped inorganic salts. It can be seen in **Figures R12-a** and **c** that *S* values of NaCl-modified membranes increase after washing, indicating the larger free volume.

On the other side, the porosity of catalytic salt (NaHCO₃) modified membrane is contributed by both the removal of trapped inorganic salts and nanobubbles of CO₂ generated by the reaction between NaHCO₃ and IP byproduct HCl. However, the later factor is believed to outweigh the former factor. NaHCO₃ in the reaction zone would be consumed by the byproduct HCl generating CO₂ as shown in **equation 1**. It can be seen that 1 mole NaHCO₃ consumed will generate 1 mole NaCl and 1 mole CO₂. **Figures R12-b** and **d** show that *S* values of NaHCO₃-modified membranes also increase after washing, indicating the increased free volume resulted from the partial removal of generated NaCl. However, the trapped salt in the PA layer can be only partially removed according to the AAS result (< 3%). Meanwhile, all generated CO₂ can create nanobubbles in the PA layer. Therefore, the higher porosity of NaHCO₃-modified membranes should be predominantly contributed by the nanobubbles of CO₂ generated.

Corresponding revision has been in the revised manuscript as follows:

“The higher porosity is ascribed to the additional nanopores originating from the removal of trapped inorganic salts for the neutral salt-modified membranes, such as NaCl@MPD. Alternatively, the removal of trapped salts and nanobubbles of CO₂

generated by reaction between catalytic salts and IP byproduct HCl^{38} are responsible for the higher porosity of catalytic salt modified membranes (such as $\text{NaHCO}_3@\text{MPD}$), where the latter might contribute predominantly.” (Pages 8-9)

Figure R12 S profiles as function of positron incident energy of salt-modified membranes before and after washing: (a) $\text{NaCl}@\text{MPD}$, (b) $\text{NaCHO}_3@\text{MPD}$, (c) $\text{NaCl}@\text{PIP}$, and (d) $\text{NaCHO}_3@\text{PIP}$

187: It is found that both MPD and TMC molecules can diffuse from the bulk aqueous phase to the water/hexane interface.

Question & Comments: How TMC diffuse from the bulk aqueous phase to the water/hexane interface?

Reply:

Thank you for your comment. We apologize for the mistake. This sentence has been corrected in the revised manuscript as follows:

“It is found that MPD molecules in both simulation systems can diffuse from the bulk aqueous phase to the water/hexane interface.” (Page 9)

273: It can be found in Figure 4b that the water permeances of modified membranes increase by ~92-165% while higher NaCl rejections well maintained (~97-99% versus ~95%) as compared to that of the pristine membrane.

Question & Comments: This rejection increase could just be the results of the increase of surface negative charges, judged by Zeta results.

Reply:

Thank you for your comment.

It is well-recognized that the increase of surface negative charges contributes to the improved salt rejection, but it is not the sole factor in this work. As stated in the manuscript, the improved salt rejection is resulted from the smaller free volume cavities with narrowly-distributed pore size and the increase of surface negative charges. The pore size distribution results in **Figure R13** obtained by the determination of neutral organic solute (no charge effect) rejection revealed that the smaller pore size also favors the improvement of solute rejection.

Figure R13 Rejection of neutral organic solutes with different molecular weights by the pristine salt-modified TFC membranes formed by PIP/TMC.

276: On the other side, the separation performance of TFC membrane fabricated by PIP and TMC was determined by NF tests. Figure 4c shows that water permeances of NaCl- and NaHCO₃-modified membranes increase by 119% and 63% respectively compared to that of the pristine membrane. Meanwhile, solute (NaCl and Na₂SO₄) rejections of modified membranes increase compared to that of the pristine membrane. Especially for NaCl rejections of modified membranes, the significant improvement of ~160-169% is achieved compared to that of the pristine membrane.

Question & Comments: The rejection of salt for PIP based membrane is normally below 40%, like pristine@PIP, but NaCl@PIP and NaHCO₃@PIP is unusually high; in fact, these membranes become loose RO membrane made by MPD, which is not good, judged by the normal function of NF membrane: separation of NaCl from Na₂SO₄. Therefore, I guess because there is a lot of temperately trapped NaCl, it repels the NaCl and other salts pass through membrane, which slows down the dissolve/diffuse process during normal separation.

Reply:

Thank you for your comment.

We agree that the trapped salts in the PA layer can repel NaCl and other salts passing through the membrane. The influence of trapped salts on the solute rejection might depend on the salt content in the PA layer. However, the trapped salt content only accounts for 0.016% of the weight of the PA layer ($< 0.09 \mu\text{g cm}^{-2}$), which might not play a significant role to determine the salt rejection. Therefore, the unusually high NaCl rejections of NaCl@PIP and NaHCO₃@PIP membranes are believed due to combined factors of smaller pore size, more negative charges, and the repel of trapped salts in the PA layer.

Additionally, salt-modified membranes, such as NH₄Cl@PIP and Na₂SO₃@PIP, show the considerably high water permeance ($25.7\text{-}38.9 \text{ L m}^{-2} \text{ h}^{-1} \text{ bar}^{-1}$) and high Na₂SO₄ rejection (94.9-96.4%), while low NaCl rejection (25.2-38.7), which can realize the normal function of NF membrane that separating NaCl from Na₂SO₄. We believe that the pore size of PIP-based NF membranes can be intentionally tuned to achieve high water permeance and Na₂SO₄ rejection, and low NaCl rejection, such as the deployment of di- or trivalent salt or salt with larger molecular weight for fabrication.

Corresponding revision has been made in the revised manuscript as follows:

“In addition, the reverse salt fluxes of modified membranes are lower than that of the pristine membrane, overcoming the permeability-selectivity tradeoff relationship. This behavior is due to the combined factors of increased negative charges and smaller free volume pore size. Additionally, the trapped salts in the PA layer repel solutes passing through the membrane, also favoring the improved solute rejection..” (Page 13)

Reviewer #2 (Remarks to the Author):

This study used a facile and versatile modification on IP process by adding different inorganic salts into the aqueous monomer solution to fabricate TFC membranes with depth-homogeneous PA layer. Molecular dynamics simulation and various characterization techniques fully demonstrated that the salt addition confines and regulates the diffusion of amine monomers to the water-oil interface and thus tunes the nanoscale-homogeneity of the PA layer. The properties of the prepared TFC membrane were improved, such as permeation flux, solute rejection and anti-fouling performance. This report is interesting and worthy to publish after a major revising. However, for the benefit of the readers and publishers, some issue should be explained and modified before being accepted. They are as followed:

Reply: Thank you for the valuable comments and suggestions that help improve our work. We have revised our manuscript based on your comments. Hopefully the revised version can meet your requirements.

1. In Lines of 145-147, “As a result,, causing a denser middle layer with thinner thickness””. In Lines of 255-258, “XPS results summarized in Table S3 and Figure 1g, indicating the lower crosslinking degree of the modified PA layers”. How does the author explain the relationship between a denser middle layer with thinner thickness and the reduced crosslinking degree of the modified PA layer?

Reply:

Thank you for your valuable comment.

Commonly, the detection depth of XPS technique for polymeric material is less than 10 nm[26], hence the detected O/N ratio actually only refers to the elemental

composition of the top skin of the PA layer, which should be the region formed in the diffusion-limited growth regime (relatively loose part). To detect the middle part of the resulting PA layers, the in-depth profile of elemental compositions was therefore further measured by XPS using $C_{24}H_{12}^+$ ion source to etch membranes. The results in **Figure R1** reveal that the modified membrane possesses a relatively loose top skin (high O/N ratio) with high Na content (**Table R2**). It was reported in previous studies that the cationic ions could complex with the carbonyl group of TMC molecules, resulting in the increased hydrolysis of acyl chloride groups¹⁵⁻¹⁷. Additionally, due to the formed denser middle layer with higher resistance, fewer amine monomers can diffuse to react with TMC in the diffusion-limited growth regime. As the result of above two factors, the modified membrane shows the lower crosslinking degree of the top skin PA layer formed in the diffusion-limited growth regime. With a deeper detection depth, the position might reach the middle part of the PA layer in the modified membrane, exhibiting lower O/N ratios (higher crosslinking degree) and Na content. In comparison with the pristine membrane, the O/N ratio of the modified membrane is even slightly lower than that of the pristine membrane, suggesting the formation of a relatively denser middle layer.

As stated in the manuscript, the presence of salt in IP process causes the reduced diffusivity of amine monomers and the shrinkage of the reaction zone. Additionally, the reduced amine monomers diffusivity results in the accelerated local rate of reaction, which might facilitate the crosslinking reaction rather than the formation of new polymers as occurred in the conventional IP process. As the result, the polymer quickly formed and accumulated in a confined reaction zone, thus the formation of a relatively denser middle layer with thinner thickness[27]. This speculation can be testified by the measured density of free standing PA films by a pycnometer (AccuPyc

1330, Micromeritics, USA) utilizing Helium as the probing gas. The density of NaCl@MPD ($1.238 \pm 0.023 \text{ g cm}^{-3}$) is slightly higher than that of Pristine@MPD ($1.183 \pm 0.017 \text{ g cm}^{-3}$), suggesting the compacter packing of PA chains in modified membranes.

Corresponding revision has been made in the revised manuscript as follows:

“Meanwhile, the in-depth O/N ratio profiles shown in **Figures 1e**, and **S5** and **Table S4** reveal that the O/N ratios of the modified membrane vary slightly against the detection depth in contrast to the fluctuating O/N ratio profiles of the pristine membrane, confirming the nanoscale homogeneity of modified membranes again. Additionally, the O/N ratio profiles also substantiate our aforementioned speculation that the middle PA layer is relatively denser.” (**Page 8**)

“This denser structure can also be verified by the smaller S values, (**Figures 1f** and **S6**) smaller d -spacing values as determined by XRD results (**Figure S7**) and higher PA intensity (**Table S5**).” (**Page 9**)

Figure R1 XPS depth profiles of membranes Pristine@MPD and NaCl@MPD; **(a)** wide-scan spectra, and **(b)** O/N ratios.

Table R2 Elemental compositions of membranes Pristine@MPD and NaCl@MPD by

XPS depth detection

Membrane code	C	O	N	Na	Cl
Pristine@MPD-0nm	72.53	16.59	10.88	/	/
Pristine@MPD-20nm	72.31	16.16	11.53	/	/
Pristine@MPD-40nm	74.03	14.95	11.02	/	/
Pristine@MPD-60nm	73.12	15.65	11.23	/	/
Pristine@MPD-80nm	73.39	15.03	11.58	/	/
NaCl@MPD-0nm	70.71	15.93	10.25	1.38	1.73
NaCl@MPD-20nm	72.40	14.56	11.13	0.73	1.18
NaCl@MPD-40nm	72.58	14.62	11.46	0.47	0.87
NaCl@MPD-60nm	72.32	14.89	11.77	0.40	0.62
NaCl@MPD-80nm	72.69	14.85	11.58	0.29	0.59

2. There are some mistakes in the marking of Figure 4(b), please correct them.

Reply:

Thank you for pointing out our mistake. We have corrected this figure as shown below.

Figure R14 Water permeance (*A*) and NaCl rejections (*R_s-NaCl*) of the pristine and modified membranes formed by MPD and TMC using 1000 ppm NaCl solution as the

feed in RO tests.

3. The contrast of (top) SEM images of modified membranes before washing in Figure S2 (a) are too small to be clear. Please place the clearer images.

Reply:

Thank you for your kind suggestion. We have adjusted the contrast of these figures to make them clearer.

Corresponding revision has been made in the revised Supporting Information.

Figure R15 SEM images (top) and corresponding Na mapping images (bottom) of modified membranes before and after washing.

4. Based on the results of Figure S1, Figure S2 and Table S1, how can the author determine that the Na atom percentage of the modified membrane after washing is all the salts trapped in the formed PA layer rather than deposited on the PSF substrate or between the substrate and PA layer?

Reply:

Thank you for your valuable comment.

We apologize for the inappropriate statement.

To detect the salt content in the PA layer, atomic absorption spectroscopy (AAS, iCE

3000, Thermo Scientific, USA) was performed for both the salt-modified membranes (10*6 cm²) and free-standing PA layers, both of them were washed by DI water for 1 day or 6 days. It can be found that the salt content trapped in the PA layer after 1-day and 6-day washing decreases slightly from 0.0902/0.0888 to 0.0871/0.0856 μg/cm² of membranes NaCl@MPD/NaHCO₃@MPD, respectively. On the other side, the content of trapped salt in the free-standing PA layer (NaCl@MPD) reduces slightly from 161.455 mg/kg to 157.280 mg/kg for 1-day and 6-day washing. The above results are consistent with the result in **Figure R8**. It can be deduced that the large amount of salt removed in the first day is generally the salt trapped in the substrate membrane as well as between the substrate membrane and the PA layer. However, after the first-day's washing, the salt trapped in the PA layer can be partially washed out with longer washing time (6 days) due to the complexation between the salt ion and the carbonyl groups of the PA network.

Corresponding revision has been made in the revised manuscript as follows:

“Eventually, due to the Lewis acid-base complexation between cations (Lewis acid) and carbonyl groups (Lewis base) of the PA network, those salts trapped in the PA layer can be partially washed out (**Figures S1-S2 and Tables S2-S3**).” (**Page 8**)

Figure R8 NaCl concentration in the washing water of MPD@NaCl membrane

5. In the Section of Method, “these membranes annealed at 80 for 15 min”. As known, soluble inorganic salts can be thermally induced crystallization. Did the author consider the effect of the generation of inorganic salts crystal on PA layer in the heat treatment process and hence affects the morphology of PA layer, or even the formation of large crystals at high inorganic salt concentration, resulting in defects of PA layer.

Reply:

Thank you for your valuable comment.

We apologize for the inappropriate statement about the heat treatment during the membrane fabrication. As reported in previous studies, TFC membranes are usually annealed in an oven to evaporate the organic solvent and complete the IP reaction to increase the crosslinking degree. However, this procedure could also result in the pore collapse of the substrate membranes. According to our experimental experiences, salt crystals can be generated on the modified membrane if drying in air or oven without washing by water as shown in **Figure R16**. Therefore, in this work, the heat treatment

was conducted by immersing the membrane sample in hot water, according to the experimental protocol reported in our previous studies[28, 29]. Thus, the salt crystals won't form on the modified membranes after heat treating by hot water (**Figure R16**). The modified membranes without these defects caused by the salt crystals was confirmed by the high solute rejection. Besides, the presence and absence of NaCl crystals on modified membranes with and without washing can also be testified by the sharp peak of XRD patterns representing NaCl crystal shown in **Figure R17**.

Corresponding revision has been made in the revised manuscript as follows:

“Finally, these membranes were posttreated in an 80 °C water bath for 15 min to eliminate the salt crystallization and then stored in DI water before use.” (**Page 17**)

Figure R16 SEM images of NaCl@MPD membranes after air drying (room temperature) without washing, drying in oven (80 °C) without washing, and immersing in hot water bath (80 °C)

Figure R17 XRD patterns of **(top)** the Pristine@MPD membrane and NaCl@MPD membrane with and without washing; **(bottom)** NaCl salt.

6. There are some tiny grammar mistakes in this manuscript draft. Please re-read it carefully and correct them.

Reply:

Thank you for your comment. We have thoroughly checked and revised the whole manuscript to eliminate these typo and grammar mistakes. Additionally, the manuscript was further polished by a professional language-editing agent (American Journal Experts) to improve the readability. The editing certificate is attached as below (**Figure R18**). We hope the revised manuscript could meet your requirement.

Figure R18 Editing certificate for the manuscript issued by a professional language-editing agent of American Journal Experts.

Reviewer #3 (Remarks to the Author):

The manuscript presents an interesting and simple way of modifying the interfacial polymerization process by adding inorganic salts into the aqueous phase. The authors report improvements in performance of the formed membranes. I think the manuscript would be acceptable for publication following a major revision as noted below:

Reply: Thank you for the valuable comments and suggestions that help improve our work. We have revised our manuscript based on your comments. Hopefully the revised version can meet your requirement.

1) The English language needs serious improvements – the manuscript is, at times, difficult to read. For example: “this result is consistent to the rejection”, “IP process undergoes a different manner”, “As the result, the locate rate of reaction accelerates”, “As the result, the resulting PA layer” and many, many other language problems which make the reception of the manuscript very difficult.

Reply:

Thank you for your comment. We have thoroughly checked and revised the whole manuscript to eliminate these typo and grammar mistakes. Additionally, the manuscript was further polished by a professional language-editing agent (American Journal Experts) to improve the readability. The editing certificate is attached as below (**Figure R18**). We hope the revised manuscript could meet your requirement.

Figure R18 Editing certificate for the manuscript issued by a professional language-editing agent of American Journal Experts.

2) In the introduction, while discussing Figure 1a, the Authors seem to suggest that generally the IP layers of TFC membranes have low salt rejections. However, commercial RO membranes have salt rejections upwards of 99% even for monovalent salts. Therefore such statements are misleading to the reader in my opinion. It is the low water permeance that is the key problem that has to be addressed or an improvement of the balance between rejection and water permeance.

Reply:

Thank you for your comment and constructive suggestion. According to your kind advice, we have revised it as follows:

“The fabrication was first developed by Cadotte in the 1970s¹¹, and has been successfully commercialized in water purification *via* nanofiltration (NF), reverse osmosis (RO) and forward osmosis (FO) processes. The separation performance of a TFC membrane is primarily governed by the characteristics of the PA active layer, including its surface and bulk properties. The design and modification of the PA layer are thus of great importance to tune the separation performance. Extensive efforts have been devoted to this end, such as facilitating amine diffusion by adding surfactant or catalyst in monomer solutions¹²⁻¹⁴, eliminating defective sites by a longer polymerization time or a higher reaction temperature¹⁵⁻¹⁸, forming crumpled Turing structures^{19,20}, deployments of molecular layer-by-layer method²¹ or electrospaying-assisted IP process²², etc^{23,24}. Despite tremendous advancements achieved *via* above strategies, researchers still cry out to the key target of bypassing the so-called permeability–selectivity trade-off to a greater extent.” (Page 3)

3) The authors say that: “scarce studies have been reported ... to finely tuning the pore size of free volume pores” in the literature. However, they fail to cite the recent

work which was dedicated precisely to addressing these issues:

a. <https://doi.org/10.1002/adma.202001132>

b. <https://doi.org/10.1016/j.memsci.2020.118572>

c. <https://patents.google.com/patent/US20190299168A1/en>

d. <https://doi.org/10.1016/j.memsci.2019.02.032>

e. <https://doi.org/10.1039/C7TA07819F>

Reply:

Thank you for your comment. We apologize for missing information about these comprehensive studies in the Introduction part. We have revised this section to include these studies as stated below:

“Extensive efforts have been devoted to this end, such as facilitating amine diffusion by adding surfactant or catalyst in monomer solutions¹²⁻¹⁴, eliminating defective sites by a longer polymerization time or a higher reaction temperature¹⁵⁻¹⁸, forming crumpled Turing structures^{19,20}, deployments of molecular layer-by-layer method²¹ or electro spraying-assisted IP process²², etc^{23,24}.” (Page 3)

“Additionally, a highly selective and permeable submicroporous TFC membrane with a large fraction of finely tuned structural submicroporosity was developed by using highly contorted triptycene building blocks of bridged-bicyclic tetra-acyl chloride as monomers³².” (Page 5)

4) I am also missing the acknowledgment that it is currently well-established in the field that heterogeneity of the IP layer structure in the TFC membranes has a significant positive impact on its properties. It is not true that flat (low roughness) and homogeneous layers are hard to achieve (see PIP-based commercial membranes) or that they have better separation properties. Very rough structures of tight RO

membranes also perform excellent in real applications. The Authors make it sound as if achieving a homogenous membrane is the Holy Grail. This is not true. In fact, achievement of a relatively thick dense layer (50 – 100 nm) is often less beneficial than highly rough crumpled structure with effectively very thin skins of the “bubbles” (~10-20 nm).

Reply:

Thank you for your comment. We agree that both the heterogeneity and rough structures of the PA layer impose positive impacts on the separation performance. Massive efforts were also dedicated to achieving rough structures, such as crumpled structure, which was proven to be of great effectiveness to increase the water permeance by providing large effective permeable area. However, the high roughness of a membrane would also compromise the antifouling property to a certain extent. Therefore, a TFC membrane with excellent separation performance and fouling resistance is preferred. As reported in the currently published paper in Science (Science 371, 72–75 (2021)), 3D models of the nanoscale PA density maps of various RO membranes constructed by Culp et al. revealed that the density fluctuations were detrimental to water transport. They suggested that controlling over the internal nanoscale inhomogeneity could maximize the water permeance by minimizing mass fluctuations without sacrificing solute rejections. Therefore, improving the nanoscale homogeneity of the PA layer is presumably to be of an effective strategy to surpass the permeability-selectivity trade-off.

In this work, the salt-modified TFC membranes also possess tuned nanoscale homogeneity as evidenced by the narrowly distributed free volume pore size and slightly varied O/N ratios (**Figure R1**), which is beneficial to the high water permeance. It is also noteworthy that except for the nanoscale homogeneity, the high

porosity and thin dense barrier (confirmed by **Figure R3**) also contribute to the high water permeance. **Figure R3** shows that the intrinsic thicknesses (PA wall) marked by red lines of modified membranes are even thinner than that of the pristine membrane. Additionally, specifically for TFC membranes formed by MPD and TMC, it can be seen by a closer observation to the surface morphology of modified membranes that the sunken thin PA films (crater) within the honey-bomb lattice (belt) cover the surface. This distinct belt-crater morphology might also benefit water permeation as the belt creates additional filtration area and crater lowers the transport resistance. Therefore, the merits of nanoscale homogeneity, high porosity and thin thickness contribute together to the greatly improved water permeance.

Corresponding revision has been made in the revised manuscript as follows:

“To solve this nerve-racking problem, deep insights should be applied to the complicated IP process to disclose the underlying mechanism. In principle,Previously, it was well-recognized that the nanoscale heterogeneity of the PA layer imposes positive impacts on the separation performance. However, 3D models of the nanoscale PA density maps of various reverse osmosis (RO) membranes constructed by Culp et al. revealed that the density fluctuations were detrimental to water transport^{29,30}. They suggested that controlling over the internal nanoscale inhomogeneity could maximize the water permeance by minimizing mass fluctuations without sacrificing solute rejections. Therefore, improving the nanoscale homogeneity of the PA layer is presumably to be of great significance in surpassing the permeability-selectivity trade-off.” **(Page 5)**

“These reported strategies in previous studies mainly aimed at reducing the PA layer thickness, tailoring the intrinsic property or increasing the effective permeable area of the PA layer. Among them, the third strategy that achieves a rough crumple structure

is proven to be of great effectiveness to improve the water permeability without sacrificing solute rejection. For a comparison, TFC membranes developed in this work not only show competitive separation performances, but also possess improved fouling resistance rather than the compromised antifouling capacity of rough structures. Therefore, the salt-tuned IP process is believed to be a promising platform for the fabrication of next-generation PA-based TFC membranes.” (Pages 15-16)

Figure R1 XPS depth profiles of membranes Pristine@MPD and NaCl@MPD: (a) wide-scan spectra, and (b) O/N ratios.

Figure R3 SEM images of the pristine and modified membranes

5) Figure 4b has a swapped legend between the permeance and rejection.

Reply:

Thank you for pointing out our mistake. The revised figure is shown as below.

Figure R14 Water permeance (A) and NaCl rejections (R_s -NaCl) of the pristine and modified membranes formed by MPD and TMC using 1000 ppm NaCl solution as the feed applied in RO separation.

6) I am missing information on how experimentally was the PALS done – as far as I am aware, PALS need a penetration depth on the scale of millimeters. Was the TFC IP layer isolated and stacked? Or was it done on powder in a “one-pot” reaction – that is not on an actual selective layer?

Reply:

Thank you for your comment. The PALS characterization was done on the TFC membrane at R & D Center for Membrane Technology at Chung Yuan University in Taiwan. Generally, the thickness for PALS characterization of dense symmetric membranes should be approximate to or larger than 1 mm, which can be the stacked

membranes[30-32] or other dense membranes[33, 34]. While for the asymmetric membranes, such as thin-film composite membranes[35-40], the membrane with thickness less than 1 mm also can be characterized by PALS technique. The evidence for the use of TFC membrane can be confirmed by the *S* profile versus positron incident energy (PIE) as shown in **Figure R19**. A reported three-layer model is applied to analyze the obtained results, i.e. (I) dense PA layer, (II) transition layer from dense layer to porous PSf support layer, and (III) porous PSf support layer. *S* parameter decreases gradually after sharp rising, demonstrating the gradual transition from the dense PA layer to the support layer (transition layer). Thereafter, *S* parameter increases slowly, corresponding to the transformation from the transition layer to the porous support layer.

The detailed experiment procedure for PALS characterization is described as below. A variable monoenergetic slow positron beam was operated in the range of 0–30 keV positron incident energy (equivalent to a mean depth of 0–10 μm) at room temperature under a vacuum of $\sim 10^{-8}$ torr. The positron source is of a 50 mCi of ^{22}Na radioisotope beam. Each membrane sample for PALS characterization was in area of $1 \times 1 \text{ cm}^2$. The as-fabricated membrane samples were measured by two positron annihilation techniques, including Doppler broadening of energy spectra (DBES) and positron annihilation lifetime (PAL) measurements. The DBES spectra were measured using a solid state HP Ge detector (EG & G Ortec) at a counting rate of approximately 3000 counts per second (cps). The total number of counts for each DBES was 1.0 million. The *S* parameter data from DBES was fitted by VEPFIT program. The PALS spectra were obtained by taking coincident events between two signals—the start signal detected by a multichannel plate from the secondary electrons and the stop signal discerned by a BaF_2 lifetime detector from the annihilation photons at a

counting rate of ~200-300 cps. A PALS spectrum contains 2.0 million counts. The positron lifetimes (τ) and intensities (I) were determined by PATFIT program, and the lifetime distribution was obtained by MELT analysis.

The corresponding revision has been made in the revised **Supporting Information**.

Figure R19 S profiles of the pristine and modified membranes as a function of the

positron incident energy for **(a)** MPD/TMC and **(b)** PIP/TMC systems.

7) How significant really is the MPD distribution difference with and without NaCl derived from MD? It seems to me that a difference between 63-78 Å and 58-78 Å zones would not make a huge difference. Also, what does “partial MPD monomers in aqueous phase” mean?

Reply:

Thank you for your valuable questions.

In fact, the scale of simulation is often much smaller than that of the real experiment condition, because of the limitation of computational resources and efficiency, leading to the non-obvious width difference of reaction zone between the pure MPD and MPD/NaCl simulation systems. However, the simulation results show that the diffusivity (D) of MPD after adding NaCl significantly decreased to less than half of the original (from $0.6 \times 10^{-9} \text{ m}^2 \text{ s}^{-1}$ to $0.23 \times 10^{-9} \text{ m}^2 \text{ s}^{-1}$ as demonstrated in **Figure 2b**), resulting in the smaller number of MPD (~ 60) in the reaction zone of MPD/NaCl system than that in the pure MPD system (~ 80) as shown in the **Figure 2e**, indicating that the number of MPD molecules diffused to the reaction zone is reduced by 25%. This difference is appreciable in the simulation scale, considering that the total number of MPD is 200. Additionally, although the difference between 63-78 Å and 58-78 Å zones is not huge, the PA layer thickness of modified membranes decreases obviously (44-67% reduction) compared to that of the pristine membrane.

We have added related information in the revised manuscript as follows:

“It can be found that the reaction zone width of the system with NaCl ($z = 63\text{-}78 \text{ \AA}$) is thinner than that of the system without NaCl ($z = 58\text{-}78 \text{ \AA}$), and the number of MPD molecules (~ 60) in the reaction zone is less than that in the system without NaCl (\sim

80), indicating a 25% reduction in the number of MPD molecules.” (Page 10)

In addition, the actual meaning of “partial MPD monomers in aqueous phase” is “some MPD monomers”. Thank again for pointing this ambiguous expression! This sentence has been revised in the manuscript as follows:

“Additionally, the accumulation behavior of Na⁺ and Cl⁻ ions can be proven by the presence of two distinct peaks, leading to the aggregation of MPD monomers in aqueous phase as verified by the MPD peak located at 20-50 Å.” (Page 10)

8) There are very little details given in the SI regarding the performance tests.

Reply:

Thank you for your comment. According to your kind suggestion, we have detailedly described the experimental protocols for the performance tests in the revised manuscript as follows:

“The separation performance of fabricated membranes was evaluated by the osmotic-driven and pressure-driven tests at room temperature. Briefly for osmotic-driven tests, DI water and 2 M NaCl applied as the feed and draw solutions respectively were recycled to the FO system by two peristaltic pumps at the flow rate of 0.3 L/min. Each fresh membrane sample was conducted under two operation modes: active layering facing draw solution (FO mode) and active layer facing feed solution (pressure retarded osmosis, PRO mode). All data were collected after 30 min stabilization for each fresh membrane sample and at least three parallel tests were repeated. The pressure-driven tests were conducted using a crossflow apparatus (Suzhou Faith Hope Membrane Technology) for RO and NF separations. In brief, the fresh membrane samples were pre-compacted at 6 bar for 1 h before data collection. The water permeance (A) was evaluated using the pure water as the feed, and the

solute rejection (R_s) was measured with 1000 ppm NaCl or Na₂SO₄ solution as the feed, under an applied pressure of 5 bar.

The osmotic-driven separation performance was evaluated by the water flux (J_v) and reverse salt flux (J_s) determined by equations (1) and (2),

$$J_v = \frac{\Delta V_o}{A_{m,o}\Delta t} \quad (1)$$

$$J_s = \frac{\Delta(C_t V_t)}{A_{m,o}\Delta t} \quad (2)$$

where ΔV_o is the volume change of the draw solution over the testing time Δt monitored by a digital balance (FX3000-GD, AND, Japan). $A_{m,o}$ is the membrane area (3.9 cm²). C_t is the salt concentration of the feed solution detected by a conductivity meter (FE30, Mettler Toledo, Switzerland).

The pressure-driven separation performance was evaluated by water permeance (A) and solute rejection (R_s) based on equations (3) and (4),

$$A = \frac{\Delta V_p}{A_{m,p} \times \Delta t \times \Delta P} \quad (3)$$

$$R_s = \left(1 - \frac{C_p}{C_f}\right) \times 100\% \quad (4)$$

where ΔV_p is the permeate volume change over the testing time, $A_{m,p}$ is the effective membrane area (7.1 cm²), ΔP is the transmembrane pressure, C_p and C_f are the solute concentrations of the permeate and the feed detected by a conductivity meter (FE30, Mettler Toledo, Switzerland).” (Pages 19-20)

9) Looking at tables S6, S7, S8 it seems to me that the separation performance in the RO and NF of the presented membranes is not drastically different to the multitude of membranes prepared in the literature, with maybe the exception of NF with NaCl.

Reply:

Thank you for your comment. The difference of water permeance of RO membrane

developed in this work is not drastically different to the multitude of membranes prepared in the literature, which is believed due to the dense and rigid structure of PA network formed by TMC and MPD. However, the data summarized in **Table S6** revealed that the membrane developed in the present work still shows considerable increment (26-480% higher) in terms of water permeance compared to those lab-made and commercial RO membranes reported in literatures while maintaining higher NaCl rejection. On the other side, NF membranes of $\text{NH}_4\text{Cl}@PIP$ and $\text{Na}_2\text{SO}_3@PIP$ show considerably high water permeances ($25.7\text{-}38.9 \text{ L m}^{-2} \text{ h}^{-1} \text{ bar}^{-1}$) while maintaining comparable Na_2SO_4 rejection (94.9-96.4%) compared to those of reported membranes. Additionally, we believe that the water permeance of PIP-based NF membrane can be intentionally improved by the deployment of di- or trivalent salt or salt with larger molecular weight.

10) How was the thickness of the IP layers determined? I cannot find any information on that in the main manuscript or in the SI. SEM –based determination is usually very difficult in TFC composites. Did the Authors attempt to isolate the layer and measure the thickness after deposition on a silicon wafer with AFM? Or with other methods?

Reply:

Thank you for your comment. The PA layer thickness value summarized in Figure 3 was roughly obtained by measuring the height of the ridge-and-valley structure based on SEM images. To more accurately measure the PA layer thickness, TEM images for these membranes were also taken as shown in **Figure R2**, and corresponding transparent thickness of the PA layer was then updated as shown in **Figure R20**.

Corresponding revision has been made in the revised manuscript as follows:

“Meanwhile, the cross-sectional TEM images in **Figure 3c** show that the

Pristine@MPD membrane exhibits both small discrete voids of nodule-like structures (lighter region) and large protuberances of leaf-like structures (darker region). Instead, belt-like structures cover the top surface of modified membranes (**Figure 3c**). On the other side, the modified PIP-TMC PA layer displays smaller nodular structures compared to that of the pristine PA layer (**Figure 3a**). Inspection of cross-sectional SEM and TEM images reveals that modified membranes show a pronounced decrease in PA layer thickness compared to that of the pristine membrane (**Figures 3b and 3c**).”

(Page 12)

Figure R2 TEM images of the pristine and modified membranes

Figure R20 PA layer thickness of the pristine and salt-modified TFC membranes based on TEM results

References

- [1] J. Jegal, S.G. Min, K.H. Lee, Factors affecting the interfacial polymerization of polyamide active layers for the formation of polyamide composite membranes, *Journal of Applied Polymer Science*, 86 (2002) 2781-2787.
- [2] A.K. Ghosh, B.-H. Jeong, X. Huang, E.M. Hoek, Impacts of reaction and curing conditions on polyamide composite reverse osmosis membrane properties, *Journal of Membrane Science*, 311 (2008) 34-45.
- [3] L. Yung, H. Ma, X. Wang, K. Yoon, R. Wang, B.S. Hsiao, B. Chu, Fabrication of thin-film nanofibrous composite membranes by interfacial polymerization using ionic liquids as additives, *Journal of Membrane Science*, 365 (2010) 52-58.
- [4] Y. Mansourpanah, K. Alizadeh, S. Madaeni, A. Rahimpour, H.S. Afarani, Using different surfactants for changing the properties of poly (piperazineamide) TFC nanofiltration membranes, *Desalination*, 271 (2011) 169-177.
- [5] J. Xiang, Z. Xie, M. Hoang, K. Zhang, Effect of amine salt surfactants on the

performance of thin film composite poly (piperazine-amide) nanofiltration membranes, *Desalination*, 315 (2013) 156-163.

[6] J. Xiang, Z. Xie, M. Hoang, D. Ng, K. Zhang, Effect of ammonium salts on the properties of poly (piperazineamide) thin film composite nanofiltration membrane, *Journal of membrane science*, 465 (2014) 34-40.

[7] J.E. Tomaschke, I.E. Ary, Interfacially synthesized reverse osmosis membranes and processes for preparing the same, in, Google Patents, 1993.

[8] J.E. Tomaschke, Interfacially synthesized reverse osmosis membranes and processes for preparing the same, in, Google Patents, 1993.

[9] J.E. Tomaschke, Interfacially synthesized reverse osmosis membrane containing an amine salt and processes for preparing the same, in, Google Patents, 1989.

[10] J.E. Tomaschke, Interfacially polymerized, bipiperidine-polyamide membranes for reverse osmosis and/or nanofiltration and process for making the same, in, Google Patents, 2002.

[11] M. Hirose, Composite reverse osmosis membrane and method for producing the same, in, Google Patents, 2004.

[12] J.-y. Koo, J.-E. Kim, W.-J. Kim, K.S. Park, Composite polyamide reverse osmosis membrane and method of producing the same, in, Google Patents, 2001.

[13] J.-y. Koo, Y.S. Yoon, N. Kim, J.-E. Kim, Composite polyamide reverse osmosis membrane and method of producing the same, in, Google Patents, 2002.

[14] J.-y. Koo, Y.S. Yoon, Composite polyamide reverse osmosis membrane and method of producing the same, in, Google Patents, 2000.

[15] B. Tang, C. Zou, P. Wu, Study on a novel polyester composite nanofiltration membrane by interfacial polymerization. II. The role of lithium bromide in the performance and formation of composite membrane, *Journal of Membrane Science*,

365 (2010) 276-285.

[16] X. Fan, Y. Dong, Y. Su, X. Zhao, Y. Li, J. Liu, Z. Jiang, Improved performance of composite nanofiltration membranes by adding calcium chloride in aqueous phase during interfacial polymerization process, *Journal of membrane science*, 452 (2014) 90-96.

[17] K. Shen, P. Li, T. Zhang, X. Wang, Salt-tuned fabrication of novel polyamide composite nanofiltration membranes with three-dimensional Turing structures for effective desalination, *Journal of Membrane Science*, 607 (2020) 118153.

[18] L.E. Peng, Z. Yao, X. Liu, B. Deng, H. Guo, C.Y. Tang, Tailoring Polyamide Rejection Layer with Aqueous Carbonate Chemistry for Enhanced Membrane Separation: Mechanistic Insights, Chemistry-Structure-Property Relationship, and Environmental Implications, *Environ Sci Technol*, 53 (2019) 9764-9770.

[19] X. Ma, Z. Yang, Z. Yao, H. Guo, Z. Xu, C.Y. Tang, Tuning roughness features of thin film composite polyamide membranes for simultaneously enhanced permeability, selectivity and anti-fouling performance, *Journal of colloid and interface science*, 540 (2019) 382-388.

[20] Z. Liu, T. Wang, D. Wang, Z. Mi, Regulating the morphology of nanofiltration membrane by thermally induced inorganic salt crystals for efficient water purification, *Journal of Membrane Science*, 617 (2021) 118645.

[21] A.S. Reddy, G.N. Sastry, Cation [$M = H^+, Li^+, Na^+, K^+, Ca^{2+}, Mg^{2+}, NH_4^+$, and NMe_4^+] Interactions with the Aromatic Motifs of Naturally Occurring Amino Acids: A Theoretical Study, *The Journal of Physical Chemistry A*, 109 (2005) 8893-8903.

[22] L. Chen, G. Shi, J. Shen, B. Peng, B. Zhang, Y. Wang, F. Bian, J. Wang, D. Li, Z. Qian, Ion sieving in graphene oxide membranes via cationic control of interlayer

spacing, *Nature*, 550 (2017) 380-383.

[23] M.F. Roberts, S.A. Jenekhe, Site-specific reversible scission of hydrogen bonds in polymers: an investigation of polyamides and their Lewis acid-base complexes by infrared spectroscopy, *Macromolecules*, 24 (1991) 3142-3146.

[24] M.F. Roberts, S.A. Jenekhe, Lewis acid complexation of polymers: gallium chloride complex of nylon 6, *Chemistry of Materials*, 2 (1990) 224-226.

[25] Z. Tan, S. Chen, X. Peng, L. Zhang, C. Gao, Polyamide membranes with nanoscale Turing structures for water purification, *Science*, 360 (2018) 518-521.

[26] J. DeVries, Surface characterization methods—XPS, TOF-SIMS, and SAM a complimentary ensemble of tools, *Journal of materials engineering and performance*, 7 (1998) 303-311.

[27] V. Freger, Kinetics of film formation by interfacial polycondensation, *Langmuir*, 21 (2005) 1884-1894.

[28] L. Shen, J. Zuo, Y. Wang, Tris(2-aminoethyl)amine in-situ modified thin-film composite membranes for forward osmosis applications, *Journal of Membrane Science*, 537 (2017) 186-201.

[29] L. Shen, L. Tian, J. Zuo, X. Zhang, S. Sun, Y. Wang, Developing high-performance thin-film composite forward osmosis membranes by various tertiary amine catalysts for desalination, *Advanced Composites and Hybrid Materials*, 2 (2018) 51-69.

[30] K. Nagai, B.D. Freeman, A.J. Hill, Effect of physical aging of poly (1-trimethylsilyl-1-propyne) films synthesized with TaCl₅ and NbCl₅ on gas permeability, fractional free volume, and positron annihilation lifetime spectroscopy parameters, *Journal of Polymer Science Part B: Polymer Physics*, 38 (2000) 1222-1239.

- [31] S. Luo, J.R. Wiegand, P. Gao, C.M. Doherty, A.J. Hill, R. Guo, Molecular origins of fast and selective gas transport in pentiptycene-containing polyimide membranes and their physical aging behavior, *Journal of Membrane Science*, 518 (2016) 100-109.
- [32] Y.-H. Huang, Q.-F. An, T. Liu, W.-S. Hung, C.-L. Li, S.-H. Huang, C.-C. Hu, K.-R. Lee, J.-Y. Lai, Molecular dynamics simulation and positron annihilation lifetime spectroscopy: Pervaporation dehydration process using polyelectrolyte complex membranes, *Journal of membrane science*, 451 (2014) 67-73.
- [33] R. Xia, X. Cao, M. Gao, P. Zhang, M. Zeng, B. Wang, L. Wei, Probing sub-nano level molecular packing and correlated positron annihilation characteristics of ionic cross-linked chitosan membranes using positron annihilation spectroscopy, *Physical Chemistry Chemical Physics*, 19 (2017) 3616-3626.
- [34] G. Dlubek, J. Stejny, M. Alam, Effect of cross-linking on the free-volume properties of diethylene glycol bis (allyl carbonate) polymer networks: a positron annihilation lifetime study, *Macromolecules*, 31 (1998) 4574-4580.
- [35] W.-C. Chao, S.-H. Huang, Q. An, D.-J. Liaw, Y.-C. Huang, K.-R. Lee, J.-Y. Lai, Novel interfacially-polymerized polyamide thin-film composite membranes: studies on characterization, pervaporation, and positron annihilation spectroscopy, *Polymer*, 52 (2011) 2414-2421.
- [36] Y.H. Huang, S.H. Huang, W.C. Chao, C.L. Li, Y.Y. Hsieh, W.S. Hung, D.J. Liaw, C.C. Hu, K.R. Lee, J.Y. Lai, A study on the characteristics and pervaporation performance of polyamide thin-film composite membranes with modified polyacrylonitrile as substrate for bioethanol dehydration, *Polymer international*, 63 (2014) 1478-1486.
- [37] S.-H. Huang, W.-S. Hung, D.-J. Liaw, C.-L. Li, S.-T. Kao, D.-M. Wang, M.D. Guzman, C.-C. Hu, Y. Jean, K.-R. Lee, Investigation of multilayer pervaporation

membrane by positron annihilation spectroscopy, *Macromolecules*, 41 (2008) 6438-6443.

[38] Y.-H. Huang, W.-C. Chao, W.-S. Hung, Q.-F. An, K.-S. Chang, S.-H. Huang, K.-L. Tung, K.-R. Lee, J.-Y. Lai, Investigation of fine-structure of polyamide thin-film composite membrane under swelling effect by positron annihilation lifetime spectroscopy and molecular dynamics simulation, *Journal of membrane science*, 417 (2012) 201-209.

[39] H. Chen, W.-S. Hung, C.-H. Lo, S.-H. Huang, M.-L. Cheng, G. Liu, K.-R. Lee, J.-Y. Lai, Y.-M. Sun, C.-C. Hu, Free-volume depth profile of polymeric membranes studied by positron annihilation spectroscopy: layer structure from interfacial polymerization, *Macromolecules*, 40 (2007) 7542-7557.

[40] Q. An, W.-S. Hung, S.-C. Lo, Y.-H. Li, M. De Guzman, C.-C. Hu, K.-R. Lee, Y.-C. Jean, J.-Y. Lai, Comparison between free volume characteristics of composite membranes fabricated through static and dynamic interfacial polymerization processes, *Macromolecules*, 45 (2012) 3428-3435.

REVIEWERS' COMMENTS

Reviewer #1 (Remarks to the Author):

author has addressed most of reviewer's questions and comments, therefore, I am OK for publication

Reviewer #2 (Remarks to the Author):

I have carefully read the feedback comments, and all the raised comments have been well resolved.

Reviewer #3 (Remarks to the Author):

To be honest, I am impressed by the amount and quality of extra work that was done by the Authors during the revision process. I accept the responses to my comments and would recommend the manuscript for acceptance.

Point-by-point responses to reviewers' comments

on

Polyamide-based membranes with structural homogeneity for ultrafast molecular sieving

(Manuscript ID: NCOMMS-21-23772A)

Reviewer #1 (Remarks to the Author):

Author has addressed most of reviewer's questions and comments, therefore, I am OK for publication.

Reply:

Thank you for your time to review our manuscript. We express our sincere thanks for your valuable comments to improve the quality of our work and also for your kind approval to accept our manuscript for publication.

Reviewer #2 (Remarks to the Author):

I have carefully read the feedback comments, and all the raised comments have been well resolved.

Reply:

Thank you for your time to review our manuscript. We express our sincere thanks for your valuable comments to improve the quality of our work and also for your kind approval to accept our manuscript for publication.

Reviewer #3 (Remarks to the Author):

To be honest, I am impressed by the amount and quality of extra work that was done

by the Authors during the revision process. I accept the responses to my comments and would recommend the manuscript for acceptance.

Reply:

Thank you for your time to review our manuscript. We appreciate your high recognition and express our sincere thanks for your valuable comments to improve the quality of our work, and also for your kind approval to accept our manuscript for publication.